# Massive fields and Wilson spools in JT gravity

Jackson R. Fliss

*Department of Applied Mathematics and Theoretical Physics, University of Cambridge,
Cambridge CB3 0WA, United Kingdom*

`jf768@cam.ac.uk`

**Abstract**

We give a prescription for minimally coupling massive matter to JT gravity with either sign of cosmological constant directly in its formulation as a topological BF theory. This coupling takes the form of a 'Wilson spool,' originally introduced in the context of three-dimensional gravity. The Wilson spool expresses the exact one-loop partition function as the integral over a Wilson loop operator. We construct the spool by considering the partition function of a massive scalar field on Euclidean $\text{dS}_2$ and on Euclidean $\text{AdS}_2$. We discuss its extension to other geometries (including the 'trumpet' and conical defect geometries) and its relation to the three-dimensional spool through dimensional reduction.

# 1 Introduction

Barring a theory of quantum gravity, complete at all energy scales, we must leverage all of the tools that gravity as a low-energy effective theory can supply us with. One feature which emerges in gravity as a local theory at low energies is diffeomorphism invariance. The gauging of this symmetry constrains the local degrees of freedom and induces long-range order: physical observables must be dressed non-locally to fixed boundaries [1] or to state-dependent features [2]. A culminating manifestation of this long-range order is the holographic principle [3, 4] encoding the information of a system in its boundary. The drastic reduction in local degrees of freedom and pushing of physical features to the boundary are features shared with systems with gapped topological order, and there is much to be leveraged from this overlap.

This overlap is exemplified in two and three dimensions where gravity can be expressed, semiclassically, as a topological gauge theory. This correspondence is strongest for two-dimensional Jackiw-Teitelboim gravity [5, 6] which can be rewritten as a non-Abelian BF theory [7]. This has led to an incredible degree of understanding of pure JT gravity with a negative cosmological constant both in canonical quantization [8–10] and as a non-perturbative gravitational path integral [11]. This progress has led in turn to concrete insights on near-horizon and near-extremal physics of black holes in higher dimensions [12]. See [13] for a thorough review. There has been much progress on a non-perturbative understanding of JT gravity with a positive cosmological constant [14–17] as well, although there also remain many puzzles (e.g. the divergent norm of the Hartle-Hawking wavefunction [18], or the interpretation of the theory as a matrix integral [17]).

An important element of any theory of quantum gravity is its coupling to matter, and JT gravity is no exception. Much interest in JT gravity stems from the general interest in dilaton theories of gravity arising from dimensional reductions of Einstein-Hilbert gravity [19]. In addition to non-trivial dilaton potentials, more realistic dimensional reductions involve matter arising from the higher-dimensional metric as well as other fields. At a practical level, the constraints of JT gravity allow it to be solved exactly even when minimally coupled to matter [20, 21] and the correlators of this theory are again given by a random matrix integral [22]. These properties make JT gravity a well controlled area for investigating the interplay between quantum gravity and quantum matter, e.g. (i) JT gravity with auxiliary gauge fields capture large quantum corrections to higher-dimensional black holes at low temperatures [12], (ii) matter alters the algebraic structure of JT gravity as a quantum theory, allowing, e.g., Hilbert spaces to be factorized [23] or subregion entropies to be defined [24].

From the perspective of gravity as a topological field theory, coupling in matter is subtle. Most obviously, matter introduces new local degrees of freedom and potentially spoils the topological character of low-dimensional gravity. Moreover, since the matter stress-tensor $T_{\mu\nu}$ couples to the inverse metric[1], the map to BF variables is highly non-linear. One insight is that while the matter action, $I_{\text{matter}}$, is clearly not topological, its path integration

$$Z_{\text{matter}}[g_{\mu\nu}] := \int \mathcal{D}\Phi\, e^{-I_{\text{matter}}[\Phi, g_{\mu\nu}]}\ , \tag{1.1}$$

can result in an, effective, topological operator inside of the gravitational path integral. This is sensible: we are "integrating out" all local degrees of freedom. This gives some initial indication that $Z_{\text{matter}}$ might have a natural (albeit non-local) description in the variables of topological field theory and making its role as a topological operator manifest. The above reasoning matches the intuition of a gauge-theory as a low-energy effective theory: the avatars of matter are line operators which are thought of as the worldlines of charged particles [26].

This idea was recently explored and developed in the context of three-dimensional gravity in [27–29]. Three-dimensional gravity also lacks local propagating degrees of freedom and accommodates a rewriting as a topological Chern-Simons theory [30, 31]. In [27–29] it was demonstrated that the one-loop determinant of minimally coupled fields results in a line operator of the Chern-Simons connections coined the *Wilson spool*. This object is roughly seen as a sum of Wilson loops wrapping cycles of the background topology arbitrarily many times.

In this paper we establish that the same is true in JT gravity. That is, the one-loop partition function of minimally coupled matter results a compact and manifestly topological expression in terms of the BF variables. This expression is still non-local but its non-locality is mild and dictated by gauge invariance: it is again the exponential of a line operator. To be specific, we consider the partition function of scalar field of mass, $m$, minimally coupled to a background metric, $g_{\mu\nu}$. Then its one-loop partition function,

---

[1]Or in the case of fermions, to the frame [25].

interpreted as an operator in the BF theory is given by

$$Z_{m^2}[g_{\mu\nu}] = \exp\left(\mathbb{W}_j[A] + \mathsf{P}(m^2)\right) \ , \tag{1.2}$$

where $\mathbb{W}_j$ is the two-dimensional Wilson spool defined as

$$\mathbb{W}_j[A] = \frac{i}{2} \int_{\mathcal{C}} \frac{d\alpha}{\alpha} \frac{\cos\alpha/2}{\sin\alpha/2} \ \mathrm{Tr}_{\mathsf{R}_j} \mathcal{P} \exp\left(\frac{\alpha}{2\pi} \oint_{\gamma} A\right) \ , \tag{1.3}$$

and $\mathsf{P}(m^2)$ is a polynomial of $m^2$. Let us unpack the physical content of the above equations. The one-form $A$ is related to the local geometry through the coframes, $e^a$, and torsionless spin connection, $\omega$, and it wraps a non-contractible cycle of the background geometry (denoted here by $\gamma$). The trace is taken over a certain highest-weight representation, $\mathsf{R}_j$, whose Casimir is determined by the mass of the scalar field and the cosmological constant, $\Lambda$, via

$$c_2(j) + \frac{m^2}{\Lambda} = 0 \ . \tag{1.4}$$

Lastly, $\alpha$ is a parameter which is integrated along a collection of contour segments, $\mathcal{C}$, appropriate for sign of the cosmological constant. We will see that this contour integration will give $\mathbb{W}_j$ a 'spooling' interpretation and will reproduce universal logarithmic divergences that exist in two dimensions. The polynomial, $\mathsf{P}(m^2)$, is dependent on the regularization of the divergent $\alpha$ integral and on the renormalization conditions imposed on $Z_{m^2}$. In what follows we will propose a specific contour regularization in which $\mathsf{P}(m^2)$ can be stated in the representation theoretic terms as

$$\mathsf{P}(m^2) = -c_2(j) - \frac{1}{4} \ . \tag{1.5}$$

Our result is a new entry into the dictionary relating bulk Wilson line operators and worldlines of massive particles in JT gravity [9, 21].

A summary of the body of the paper follows. In Section 2 we review basics of JT gravity and its rewriting as a BF theory. We will introduce the matter theory and discuss the quasinormal mode method developed in [32] for calculating its one-loop determinant. We will reformulate this method in terms of representation theory using the principles established in [29]. In Section 2.1 we will specialize to Euclidean dS$_2$, investigate its on-shell solution, and construct the Wilson spool directly. This construction will make use of a series of non-standard representations of $\mathfrak{su}(2)$ introduced in [33] and developed in [27, 29], which we will review. We will then verify that our result reproduces a known result for the $S^2$ partition function of a massive scalar field. In Section 2.2 we will repeat this construction for Euclidean AdS$_2$ and show that our result correctly reproduces the one-loop partition function of a massive scalar on the hyperbolic disc. In Section 3 we summarize and discuss our results in a broader context. We will make note of effects coming from dynamical boundaries and higher topological configurations. In particular we express the spool on the 'trumpet' geometry that plays a key role in topological recursion, as well as on off-shell geometries possessing conical deficits (the 'cone' and the 'football').

Lastly, we discuss the possibility (and difficulties) of realizing the two-dimensional Wilson spool as a dimensional reduction from three-dimensions. Conventions and additional details are contained the Appendices.

## 2 The Wilson spool in JT gravity

**JT gravity and BF theory**

We set the stage by introducing Euclidean JT gravity with action

$$I_{\text{JT}}[\varphi, g] = -\frac{\varphi_0}{4G_N}\chi - \frac{1}{16\pi G_N}\int_M d^2x\,\sqrt{g}\,\bar{\varphi}(R - 2\Lambda) \ . \tag{2.1}$$

Above $\chi$ is the Euler characteristic of the manifold, $M$, which for a genus $\mathsf{g}$ manifold with $\mathsf{n}$ boundaries is

$$\chi = 2 - 2\mathsf{g} - \mathsf{n} \ , \tag{2.2}$$

and $\Lambda$ is the cosmological constant which defines a length scale, $\Lambda = \pm\ell^{-2}$, depending on sign. We have ignored potential boundary terms however we will introduce these later below as needed.

This is a particular instance of dilaton gravity which are common in dimensional reductions of Einstein gravity in higher dimensions. Here the dilaton, $\varphi = \varphi_0 + \bar{\varphi}$ comes with a linear potential, $U(\varphi) = -2\Lambda(\varphi - \varphi_0)$. We will not integrate over the constant, background value of the dilaton, $\varphi_0$, which multiplies the Euler characteristic, and instead regard it as a topological coupling constant suppressing higher genus configurations within the JT path integral. The path integration over fluctuating dilaton, $\bar{\varphi}$, imposes

$$R = 2\Lambda \ , \tag{2.3}$$

exactly as a constraint. Thus the JT path-integral, as stated above, appears as a sum over topological configurations of constant curvature:

$$Z_{\text{JT}} = \int \mathcal{D}\bar{\varphi}\,\mathcal{D}g_{\mu\nu}\,e^{-I_{\text{JT}}} \sim \sum_{\text{topologies}} e^{\frac{\varphi_0}{4G_N}\chi} \tag{2.4}$$

This is only partially true: for configurations with a cutoff boundary, such as those in the "nearly AdS$_2$" context, there are dynamical boundary modes which are the Nambu-Goldstone bosons for reparameterizations of the boundary coordinate [34, 35]. These dynamics are governed by a Schwarzian theory which has played a central role in duality between nearly AdS$_2$ and the Sachdev-Ye-Kitaev (SYK) model [36–38] at low energies, as well as the exact rewriting of the non-perturbative $Z_{\text{JT}}$ as a matrix integral [11]. In this paper we will be primarily concerned with the interplay of bulk physics with matter and we will leave a discussion boundary dynamics for Section 3.

Because (2.1) consists of constraints which remove all local degrees of freedom, it is expected that it can be expressed as a topological field theory. This theory is a non-Abelian BF theory with action

$$S_{\text{BF}} = \frac{k}{2\pi} \int_M \text{Tr}\,(BF)\ , \qquad F = \text{d}A + A \wedge A\ , \tag{2.5}$$

which, up to boundary terms, describes JT gravity with both signs of cosmological constant and in both Lorentzian and Euclidean signature. In Lorentzian signature $B$ and $A$ are valued in $\mathfrak{sl}(2,\mathbb{R})$ and related to the metric variables in the first order formulation, namely the coframes, $\{e^a\}_{a=0,1}$, and the spin connection, $\omega$, as

$$A = e^a\,\mathcal{J}_a + \omega\,\mathcal{J}_2\ , \qquad B = \lambda^a\,\mathcal{J}_a + \varphi\,\mathcal{J}_2\ , \tag{2.6}$$

for an appropriate basis of generators, $\{\mathcal{J}_A\}_{A=0,1,2}$, where $\mathcal{J}_a$ can be thought of generating translations and $\mathcal{J}_2$ as generating boosts [39]. The trace in (2.5) is taken in the fundamental representation.[2] The corresponding Euclidean descriptions will depend on details of the Wick rotation and the sign of the cosmological constant. We will be more specific for each case, however the details of this can be found more fully in Appendix B. Similar to (2.1), this action is purely a constraint on the local degrees of freedom: the integration over $B$ forces $A$ to be a flat connection, $F = 0$. In terms of the metric variables, $\lambda^a$ constrain torsion to vanish while $\bar{\varphi}$ will impose the curvature constraint (2.3).

At this point, it is necessary for us to comment on one key difference between JT gravity (as a theory of a metric and a dilaton) and BF theory as quantum theories. In principle, given a background topology, there are more flat connections satisfying $F = 0$ than there are invertible metrics. One immediate example is the trivial connection $A = 0$ which clearly results in a degenerate frame through (2.6). In this paper we will skirt this issue and always work about flat connections that map to geometric solutions.

**Minimally coupled matter**

The rewriting $I_{\text{JT}} = -iS_{\text{BF}}$ makes manifest that this is a theory of long-range degrees of freedom. However this manifest topological invariance is potentially spoiled through the introduction of bulk matter which couples locally to the metric. For instance a minimally coupled scalar field, $\Phi$, of mass $m$ will have action

$$I_{\text{matter}}[\Phi, g_{\mu\nu}] = \frac{1}{2} \int d^2x \sqrt{g}\,\left(g^{\mu\nu}\partial_\mu\Phi\partial_\nu\Phi + m^2\Phi^2\right)\ . \tag{2.7}$$

This action (2.7) is clearly not topological as $\Phi$ has dynamical local degrees of freedom. Moreover, since (2.7) involves the inverse metric, it clearly does not admit a local polynomial expression in terms of $A$ and $B$. The upshot of this paper that while $I_{\text{matter}}$ is not topological, the path integration over $\Phi$,

$$Z_{m^2}[g_{\mu\nu}] = \int \mathcal{D}\Phi\, e^{-I_{\text{matter}}[\Phi, g_{\mu\nu}]}\ , \tag{2.8}$$

---

[2]From here onward, "Tr" always denotes the fundamental representation. Traces in alternative representations will be explicit denoted as such, e.g. "Tr$_{\text{R}}$".

results in a topological operator that is given as the exponential of a line operator of $A$

$$Z_{m^2}[g_{\mu\nu}] = e^{-c_2(j)-\frac{1}{4}} e^{\mathbb{W}_j[A]} , \qquad (2.9)$$

with $\mathbb{W}_j[A]$ given by (1.3). In the rest of this section we will establish this result and test in the simplest on-shell geometries for both signs of cosmological constant.

**The DHS method**

The primary methodology we will use is a variant of the method of quasinormal modes established by Denef, Hartnoll, and Sachdev (DHS) [32]. In words, we continue $Z_{m^2}$ as a meromorphic function in the complex mass plane. As a meromorphic function, it is fixed (up to a total holomorphic function) as the rational product over its zeros and poles. For scalar theories, these poles correspond to spectra of quasinormal modes. The logarithm of the partition function is then given a sum over quasinormal modes which can then be neatly expressed as spectral zeta functions up to polynomial expressions of the mass.

For our purposes we will use a variation of this method established in [27, 29] which is tailored to the representation theory of local isometry algebra. In this variant, the sum over quasinormal modes is replaced with a sum over weights of a representation subject to several conditions, which are summarized below.

- **Condition 0 or the "mass shell" condition**: The representations contributing poles to $Z_{m^2}$ will have a quadratic Casimir value that is proportional to $m^2$. This condition ties the analytic continuation in the mass complex mass plane into a continuation in the representation weight plane.

- **Condition I**: A weight of a representation satisfying **Condition 0** can contribute a pole if it corresponds to a single-valued field. This in turn implies its parallel transport along any closed cycle must be trivial. This condition ties the placement of poles in the complex weight plane to the values of the holonomies of the background connection.

- **Condition II**: Weights can contribute a pole only if they correspond to globally regular solutions. When the isometry algebra exponentiates to a group which acts transitively on the background geometry, this implies that such weights belong to faithful unitary representations of this group.

While **Condition 0** is a local condition (the statement that poles correspond to mass-shell solutions), **Conditions I & II** are statements about global boundary conditions. We emphasize that the satisfaction of all three conditions do not correspond to *physical* solutions of the field or *physical* values of the mass, but instead conditions on the analytic structure of $Z_{m^2}$ within the complex mass plane. Indeed, by the definition of a pole, the satisfaction of all three conditions corresponds to $Z_{m^2} = \infty$ which is not physical at all!

Below we will utilize this formulation of the DHS method to build the Wilson spool both the Euclidean $dS_2$ and Euclidean $AdS_2$ backgrounds. We will begin with the construction on Euclidean $dS_2$ where both the representation theory and the interplay between **Conditions 0, I, & II** are more involved. The construction for Euclidean $AdS_2$ will then readily follow.

## 2.1 Euclidean dS$_2$

We will begin with a positive cosmological constant, where we can represent Euclidean JT gravity as a BF theory with fields valued in the algebra $\mathfrak{su}(2)$:

$$S_{\text{BF}} = \frac{k}{2\pi} \int_M \text{Tr}\, BF \ . \tag{2.10}$$

There is no boundary term as we will be interested in this theory defined on compact two-manifolds.

In this case the map to metric variables is given by

$$A = i \left( e^0/\ell L_1 + e^1/\ell\, L_2 + \omega L_3 \right) \ , \qquad B = i \left( \lambda^0 L_1 + \lambda^1\, L_2 + \varphi\, L_3 \right) \ , \tag{2.11}$$

where $\{e^a\}_{a=0,1}$ are Euclidean coframes, $\omega = \frac{1}{2}\varepsilon^{ab}\omega_{ab}$ is the only independent component of the spin-connection, and $\{L_A\}_{A=1,2,3}$ generate $\mathfrak{su}(2)$.[3] Under this map we have

$$iS_{\text{BF}} = \frac{1}{16\pi G_N} \int d^2x \sqrt{g}\, \varphi \left( R - \frac{2}{\ell^2} \right) + \frac{1}{8\pi G_N} \int \delta_{ab}\lambda^a T^a = -I_{\text{JT}} \ , \tag{2.12}$$

with $\lambda^a$ as Lagrange multipliers constraining the torsion to vanish:

$$T^a = \mathrm{d}e^a + \varepsilon^a{}_b\, \omega \wedge e^b = 0 \ , \tag{2.13}$$

and the BF level is related to the Newton's constant via

$$k = \frac{i}{2G_N} \ . \tag{2.14}$$

See Appendix B.2 for details.

The dilaton enforces a positive curvature constraint, $R = \frac{2}{\ell^2}$, the simplest solution to which is a round two-sphere with metric and on-shell dilaton given by

$$\frac{\mathrm{d}s^2}{\ell^2} = \mathrm{d}\rho^2 + \sin^2\rho\, \mathrm{d}\tau^2 \ , \qquad \bar{\varphi} = \sin\rho \sin\tau \ , \tag{2.15}$$

with $\rho \in [0, \pi]$ and $\tau \sim \tau + 2\pi$. As explained further in Appendix B.2, this solution can be arrived at from Wick rotation from the Lorentzian static path; in the Lorentzian coordinates $\rho = 0, \pi$ mark the locations of two separate cosmological horizons.

The corresponding flat connection is given by

$$A = i \left( \sin\rho\, \mathrm{d}\tau\, L_1 + \mathrm{d}\rho\, L_2 + \cos\rho\, \mathrm{d}\tau\, L_3 \right) = g_\rho^{-1} g_\tau^{-1} \mathrm{d} \left( g_\tau\, g_\rho \right) \tag{2.16}$$

with

$$g_\rho = e^{i\rho L_2} \ , \qquad g_\tau = e^{i\tau L_3} \ . \tag{2.17}$$

This emphasizes that $A$ is pure gauge and is locally flat almost everywhere on the $S^2$. However, a careful analysis reveals that it has two point sources of flux at the North and

---

[3]Our conventions for $\mathfrak{su}(2)$ are given in Appendix A.

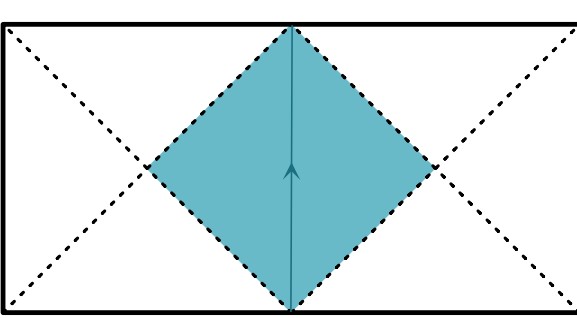
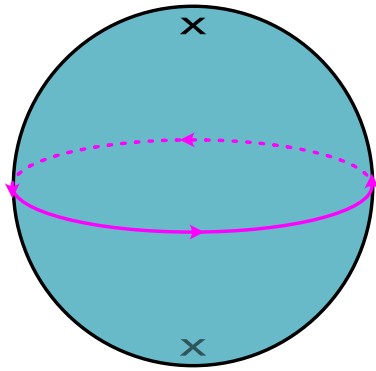

**Figure 1: Left:** The Penrose diagram for Lorentzian dS$_2$. The static patch is highlighted in blue with its horizons denoted by the dashed lines. The worldline of the timelike observer defining the patch is depicted in green. **Right:** The Euclidean geometry is a round $S^2$. The background connection possesses flux at the North and South poles which are the locations of the static patch horizons after Wick rotation. These fluxes give the Wilson loop wrapping the equator (depicted in pink) a holonomy.

South poles of the $S^2$ which are the origins of the $\tau$ coordinate. This is depicted in Figure 1. Indeed regarding, distributionally,

$$\mathrm{dd}\tau = (\delta(\rho) - \delta(\rho - \pi))\,\mathrm{d}\tau \wedge \mathrm{d}\rho \;, \tag{2.18}$$

we find

$$F = i\,(\delta(\rho) + \delta(\rho - \pi))\,\mathrm{d}\tau \wedge \mathrm{d}\rho \, L_3 \;, \tag{2.19}$$

and so the on-shell action is given by the sum of the dilaton values at the cosmological horizons:

$$iS_{\mathrm{BF,\ on\text{-}shell}} = -i\frac{k}{2}\,(\varphi(0) + \varphi(\pi)) = \frac{\varphi_0}{2G_N} \;. \tag{2.20}$$

An important effect of these fluxes is that they prevent the cycle, $\gamma$, wrapping the equator of the $S^2$ from being contractible and $A$ possesses a holonomy along this cycle:

$$\mathcal{P}\exp\oint_\gamma A = g_\rho^{-1} e^{i2\pi\mathsf{h}L_3} g_\rho \;, \qquad \mathsf{h} = -1 \;. \tag{2.21}$$

For all finite dimensional representations of $SU(2)$, this implies that the group element $\mathcal{P}\exp\oint A$ conjugates to $\pm 1$, however this will not be the case for the infinite dimensional, *non-standard*, representations of $\mathfrak{su}(2)$ introduced in [33] and developed in [27, 29]. We now explain how these representations play an important in encoding the physics of massive particles propogating on dS$_2$.

The single-particle states of a field are intimately tied to the irreducible representation theory of the background isometry group. This representation theory is also the natural language to discuss the coupling of a field to a gauge theory taking values in the isometry algebra. We focus on our scalar field, $\Phi$, minimally coupled to the metric through the

action (2.7). Its partition function on Euclidean dS$_2$ is given by

$$Z_{m^2} = \det \left( \nabla^2_{S^2} - m^2 \ell^2 \right)^{-1/2} .$$

(2.22)

As mentioned at the beginning of this section, we will evaluate this partition function in a group theoretic variant of the DHS or "quasinormal mode" method. This method relies on continuing $Z^2_{m^2}$ as a meromorphic function in the complex mass plane and equating it (up a holomorphic pre-factor) to the product of its poles.

We first notice that $S^2$ possesses an isometry group of Killing vectors, $\{\zeta_A\}_{A=1,2,3}$, realizing the $\mathfrak{su}(2)$ algebra acting on scalar functions,

$$[\zeta_A, \zeta_B] = i\epsilon_{ABC}\zeta_C ,$$

(2.23)

and whose quadratic Casimir yields the $S^2$ Laplacian[4]:

$$\delta^{AB}\zeta_A\zeta_B = -\nabla^2_{S^2} .$$

(2.25)

Thus $Z_{m^2}$ possesses poles for states of a scalar representations satisfying

$$c_2^{\mathfrak{su}(2)} + m^2\ell^2 = 0 .$$

(2.26)

This is **Condition 0**, or the "mass-shell" condition mentioned above. This condition is a physical condition tying an analytic continuation in the mass to analytic continuations in the weight spaces of $\mathfrak{su}(2)$ representations.

It is standard procedure to construct representations of $\mathfrak{su}(2)$ starting from a highest weight state satisfying

$$L_3|j,0\rangle , \qquad L_+|j,0\rangle = 0 ,$$

(2.27)

and acting freely with $L_-$. The Casimir of such representation, $\mathsf{R}_j$, is given by

$$c_2^{\mathfrak{su}(2)}|j,0\rangle = j(j+1)|j,0\rangle .$$

(2.28)

For generic masses, **Condition 0** seems to preclude any finite dimensional highest-weight representations which require $j \in \frac{\mathbb{N}}{2}$. However it was demonstrated in [29] (building off [27, 33]) that *non-standard* highest-weight representations can be constructed for any $j \in \mathbb{C}$. These representations are infinite-dimensional yet can be equipped with a positive definite inner-product, making them unitary representations of the $\mathfrak{su}(2)$ algebra[5]. Non-standard representations have well-defined characters given by

$$\chi_j(z) := \mathrm{Tr}_{\mathsf{R}_j}\left(e^{i2\pi z\, L_3}\right) = \frac{e^{i\pi z(2j+1)}}{2i\sin(\pi z)} .$$

(2.29)

---

[4]Explicitly, in the coordinates of (2.15),

$$\zeta_1 = -\,i\cos\tau\,\partial_\rho + i\cot\rho\sin\tau\,\partial_\tau ,$$
$$\zeta_2 = i\sin\tau\,\partial_\rho + i\cot\rho\,\cos\tau\,\partial_\tau ,$$
$$\zeta_3 = i\partial_\tau .$$

(2.24)

[5]Although, via the Peter-Weyl theorem, they do not exponentiate to unitary representations of the group $SU(2)$. This fact will play a role in **Condition II** shortly below.

The mass-shell condition states then states that $Z_{m^2}$ will have poles for each weight of a non-standard representation satisfying

$$j = -\frac{1}{2} \mp \sqrt{\frac{1}{4} - m^2 \ell^2} \ . \tag{2.30}$$

For definiteness we denote $j$ as the highest-weight with a '$-$' sign and $\bar{j}$ with the '$+$' sign. For 'light' fields satisfying $4m^2\ell^2 < 1$ the poles lie on 'complementary type' representations with $j = -\frac{1}{2} + \sigma$ and $\sigma \in \left(0, \frac{1}{2}\right)$, while 'heavy fields' with $4m^2\ell^2 > 1$ lie in 'principal type' representations with $j = -\frac{1}{2} - i\mu$ and $\mu \in (0, \infty)$. In this paper we will focus on heavy fields with

$$\mu := \sqrt{m^2\ell^2 - \frac{1}{4}} \ . \tag{2.31}$$

The results for light fields can then be obtained from the analytic continuation $\mu \to -i\sigma$.

We now consider **Conditions I & II** which account for the boundary conditions encoded in the functional determinant appearing in (2.22). **Condition I** is a condition that a solution contributing a pole to $Z_{m^2}^2$ must be single-valued. This corresponds to the condition that the parallel transport of $\Phi$ along any closed cycle, $\gamma$, must be trivial:

$$\Phi = \mathsf{R}_j \left[ \mathcal{P} \exp \oint_\gamma A \right] \Phi \ . \tag{2.32}$$

If $A$ possesses holonomy, $\mathsf{h}$, along this cycle then a state of $\mathsf{R}_j$ with fixed $L_3$ weight, $\lambda$, can contribute a pole when

$$\lambda \mathsf{h} \in \mathbb{Z} \ . \tag{2.33}$$

As we saw above, the background connection relevant to Euclidean dS$_2$ given in (2.16) possesses holonomy, $\mathsf{h} = -1$, for the cycle homotopic to the equator and wrapping the fluxes at the North and South poles.

Lastly, **Condition II** states that a weight can contribute a pole to $Z_{m^2}^2$ if it corresponds to a globally regular solution on $S^2$. Such solutions are given by finite dimensional representations of $SU(2)$ with weights

$$\lambda = j - p \ , \qquad p = 0, 1, \ldots, 2j \ , \qquad j \in \frac{\mathbb{N}}{2} \ . \tag{2.34}$$

We reemphasize the point made at in the introduction of this section that **Condition II** only locates the poles of $Z_{m^2}^2$ and *not* the physical values of representation weights which for real masses correspond to the non-standard representations.

The important input of **Condition II** is that finite dimensional representations are double-sided, meaning that for any weight satisfying $\lambda \mathsf{h} = n$ there is another weight satisfying $\lambda \mathsf{h} = -n$. Thus for the purposes of tracking the location and degeneracy of poles of $Z_{m^2}^2$ it is sufficient to combine these into **Condition I'**:

$$\lambda \mathsf{h} = |n| \ , \qquad n \in \mathbb{Z} \ . \tag{2.35}$$

We now put this together. Up to a holomorphic function, now regarded as a function of the highest weight, $e^{\mathsf{P}(\mu)}$, (which we will fix below), $Z_{m^2}^2$ is given by the product over simple poles in the representation weight plane:

$$Z_{m^2}^2 = e^{\mathsf{P}(\mu)} \left[ \prod_{\lambda \in \mathsf{R}_j} \prod_{n \in \mathbb{Z}} (|n| - \lambda \mathsf{h})^{-1} \right] \times \left[ \prod_{\bar{\lambda} \in \mathsf{R}_{\bar{j}}} \prod_{\bar{n} \in \mathbb{Z}} \left( |\bar{n}| - \bar{\lambda} \mathsf{h} \right)^{-1} \right] . \tag{2.36}$$

The spool construction of $\log Z_{m^2}$ now readily follows. We implement the logarithm with a Schwinger parameter

$$\log M = - \int_{\times}^{\infty} \frac{d\alpha}{\alpha} e^{-\alpha M} , \tag{2.37}$$

where "$\int_{\times} d\alpha$" denotes an $\epsilon$ regularization of the $\alpha \sim 0$ UV divergence. We will discuss this regularization in more depth below but for now we will be agnostic about its specific implementation.

The sums over $e^{-\alpha |n|}$ and $e^{-\alpha |\bar{n}|}$ are geometric and easily performed and we further recognize the sum over weights of $e^{\alpha \lambda \mathsf{h}}$ as the representation trace over a Wilson loop of $A$. All-in-all this leads to

$$\log Z_{m^2} = \frac{1}{2} \int_{\times}^{\infty} \frac{d\alpha}{\alpha} \frac{\cosh \alpha/2}{\sinh \alpha/2} \left[ \mathrm{Tr}_{\mathsf{R}_j} \mathcal{P} \exp\left( -i\frac{\alpha}{2\pi} \oint_{\gamma} A \right) + \mathrm{Tr}_{\mathsf{R}_{\bar{j}}} \mathcal{P} \exp\left( -i\frac{\alpha}{2\pi} \oint_{\gamma} A \right) \right] , \tag{2.38}$$

up to a polynomial, $\mathsf{P}(\mu)$. Lastly we change integration variables, $\alpha \to i\alpha$, and noticing that integrand is odd in $\alpha$, the sum over representations is exchanged for a sum of integration contours. This leads our main result:

$$\log Z_{m^2} = \mathbb{W}_j[A] + \mathsf{P}(\mu) \tag{2.39}$$

with

$$\mathbb{W}_j[A] = \frac{i}{2} \int_{\mathcal{C}} \frac{d\alpha}{\alpha} \frac{\cos \alpha/2}{\sin \alpha/2} \mathrm{Tr}_{\mathsf{R}_j} \mathcal{P} \exp\left( \frac{\alpha}{2\pi} \oint_{\gamma} A \right) , \tag{2.40}$$

where $\mathcal{C}$ is the union of two contour segments, $\mathcal{C}_{\pm}$ which run along the imaginary $\alpha$ axis, from an infinitesimal distance of the origin, $i0^{\pm}$, and directed toward $\pm i\infty$, as depicted in Figure 2.

Let us make some comments about this result.

- The two-dimensional Wilson spool, (2.40), takes a form familiar to the result established in [27], namely as the integral over a topological line operator. This is in keeping with the interpretation of the Wilson loop as the low-energy avatar of a massive particle world-line. Additionally the measure multiplying this line operator remains unchanged: as we will soon explain, this ensures an (albeit looser) interpretation of "spooling" Wilson loops around the $S^2$ equator.

  One stark difference between our result and that established in [27] lies in the contour, $\mathcal{C}$. Unlike the three-dimensional spool which is integrated over complete closed

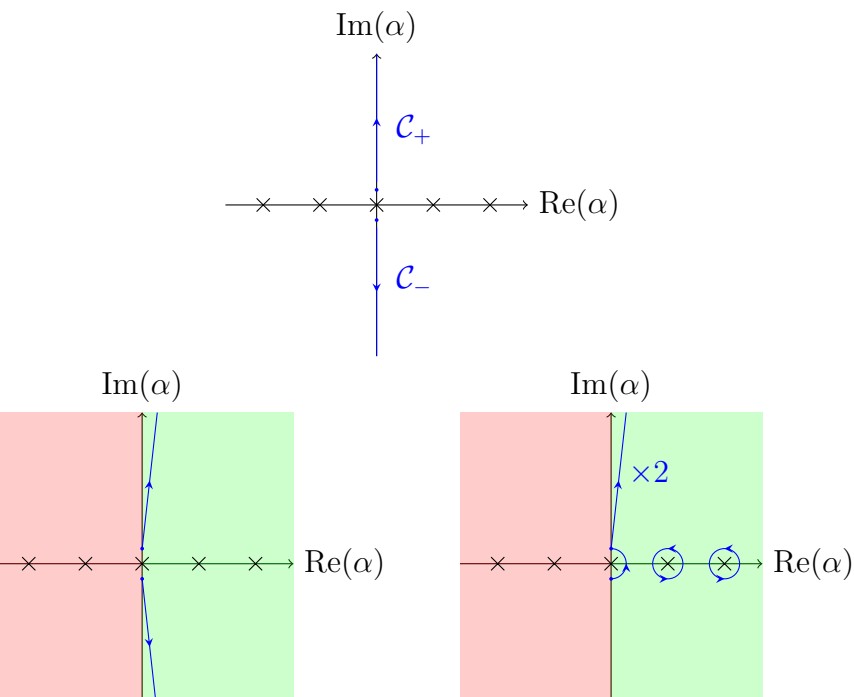

**Figure 2: Top:** The integration contour segments $\mathcal{C}_{\pm}$ running in opposite directions from infinitesimally outside $i0^{\pm}$ to $\pm i\infty$. The poles of the on-shell integrand are depicted as crosses at $\alpha \in 2\pi\mathbb{Z}$. **Bottom left:** One deformation of the integration contours into the damped region (shaded in green). The anti-damped region is shaded in red. **Bottom right:** A possible rotation of the integration contours, which include contributions from the half-contour around the origin as well as the residues of poles at $\alpha \in 2\pi\mathbb{N}$.

contours in the $\alpha$ plane, the integration in (2.40) consists of two open contours emanating in opposite directions from the origin cannot be completed into a single closed contour.

- A direct consequence of the peculiar contour in (2.40) is that there is no preferred way to regulate the UV divergence as $\alpha \to i0^{\pm}$. As a result, $\mathbb{W}_j$, will remain sensitive to this regulator. However this is physically expected: in even dimensions $\log Z_{m^2}$ will possess a logarithmic divergence which is universal, i.e. it cannot canceled be local counterterms. Simple residues of the integrand of (2.40) will never result in such a logarithm; in order to capture this logarithmic divergence, $\mathbb{W}_j$ must remain sensitive to how we approach $\alpha \sim i0^{\pm}$. We will verify shortly that $\mathbb{W}_j$ possesses both the expected power law and logarithmic divergences.

- The exact form of the polynomial contribution, $\mathsf{P}(\mu)$, is dependent both on the regularization scheme and the renormalization condition by which one removes the local divergences from $\log Z_{m^2}$. Below will fix it with a specific regulated contour and comparing to the minimally subtracted $\log Z_{m^2}$ obtained by heat kernel.

- Despite the fact that $\mathcal{C}$ no longer consists of closed contours, we can still give $\mathbb{W}_j$

a spooling interpretation. As emphasized in [40], the contour segments in (2.40) actually lie along Stokes lines of the $\alpha$ integrand. In order to properly define this integral we should rotate rotate both $\mathcal{C}_\pm$ into a region where the integrand is damped. In doing so, we can choose to rotate one of the integration contours through to wrap the poles at $\alpha = 2\pi n$ ($n \in \mathbb{N}$), as depicted in Figure 2. Analogous to the result in three-dimensions, this sum is the avatar of particle worldlines wrapping the $S^2$ arbitrarily many times.

**On-shell test**

We now unravel our result and show that, when taking the connection on-shell as (2.16), it reproduces the correct one-loop determinant. We firstly note from (2.21) that the Wilson loop of $A$ wrapping the equator possesses holonomy $\mathsf{h} = -1$. The on-shell value of the Wilson loop appearing in (2.40) is the representation character

$$\mathrm{Tr}_{\mathsf{R}_j} \mathcal{P} \exp\left( \frac{\alpha}{2\pi} \oint_\gamma A \right) = \chi_j\left( -\frac{\alpha}{2\pi} \right) = \frac{i}{2} \frac{e^{-\alpha\mu}}{\sin(\alpha/2)} \ . \tag{2.41}$$

Taking our integration back to a real variable, $\alpha \to -i\alpha$, we find that the Wilson spool exactly reproduces the integral form of the (unregulated) one-loop determinant of a massive scalar field on the two-sphere [41][6]:

$$\mathbb{W}_j = \frac{1}{2} \int_{\times}^{\infty} \frac{\mathrm{d}\alpha}{\alpha} \frac{\cosh \alpha/2}{\sinh^2 \alpha/2} \cos(\alpha\mu) = \log Z_{m^2}[S^2] - \mathsf{P}(\mu) \ . \tag{2.42}$$

We have been agnostic about the regularization scheme (and thus specifying $\mathsf{P}(\mu)$), but for definiteness we will compute the regulated partition function by cutting off our contour segments:

$$\int_{\times}^{\infty} \mathrm{d}\alpha := \lim_{\epsilon \to 0} \int_{\epsilon}^{\infty} \mathrm{d}\alpha \ . \tag{2.43}$$

This computation is performed explicitly in Appendix C to find

$$\begin{aligned}
\mathbb{W}_j &= \frac{1}{\epsilon^2} + \left( \mu^2 - \frac{1}{12} \right) \log(e^{\bar{\gamma}}\epsilon) + \sum_{\pm} \left[ \zeta'\left( -1, \frac{1}{2} \pm i\mu \right) \mp i\mu\zeta'\left( 0, \frac{1}{2} \pm i\mu \right) \right] \ , \\
&= \frac{1}{\epsilon^2} + \left( \mu^2 - \frac{1}{12} \right) \log\left( -e^{\bar{\gamma}}\epsilon \right) + 2\zeta'\left( -1, \frac{1}{2} + i\mu \right) - 2i\mu\zeta'\left( 0, \frac{1}{2} + i\mu \right) \\
&\quad + i\mu \mathrm{Li}_1\left( -e^{-2\pi\mu} \right) + \frac{i}{2\pi} \mathrm{Li}_2\left( -e^{-2\pi\mu} \right) \ , \tag{2.44}
\end{aligned}$$

where $\zeta(z, a)$ is the Hurwitz zeta function (the prime indicates a derivative with respect to $z$), $\bar{\gamma}$ is the Euler-Mascheroni constant, and

$$\mathrm{Li}_q(z) = \sum_{n=1}^{\infty} \frac{z^n}{n^q} \ , \tag{2.45}$$

---

[6]See, e.g., equation (3.10) in that paper. For comparison, in our conventions $\mu_{\mathrm{us}} = \nu_{\mathrm{them}}$.

are polylogarithms. In the first equality we evaluate $\mathbb{W}_j$ along the contour segments depicted in the bottom left of Figure 2 which emphasizes the reality of the answer. In the second equality we perform the integral along the contours depicted on the bottom right of Figure 2; the polylogarithms indicate the presence of a 'winding sum' of Wilson loops. Nevertheless both expressions are equivalent.

Upon minimal subtraction, our result, (2.44), differs from that of the minimally subtracted heat kernel[7] by a polynomial term which fixes

$$\mathsf{P}(\mu) = \mu^2 = - \left( c_2^{\mathfrak{su}(2)}(j) + \frac{1}{4} \right) \ . \tag{2.46}$$

## 2.2 Euclidean AdS$_2$

We now move to Euclidean JT gravity with negative cosmological constant which can be expressed as an $\mathfrak{sl}(2, \mathbb{R})$ BF theory,

$$S_{\text{BF}} = \frac{k}{2\pi} \int_M \text{Tr}\,(BF) - \frac{k}{4\pi} \oint_{\partial M} \text{Tr}(BA) \ . \tag{2.47}$$

This equivalent to the usual JT action through writing

$$A = -\left( e^a\, P_a + \omega P_2 \right) \ , \qquad B = -i\left( \lambda^a\, P_a + \varphi\, P_2 \right) \ , \qquad (a = 0, 1) \ , \tag{2.48}$$

where $\{P_A\}_{A=0,1,2}$ generate $\mathfrak{sl}(2, \mathbb{R})$[8]. Then (2.47) takes the form of the Euclidean JT action with negative cosmological constant,

$$\begin{aligned}
iS_{\text{BF}} &= \frac{1}{16\pi G_N} \int d^2x \sqrt{g}\varphi \left( R + \frac{2}{\ell^2} \right) + \int \lambda^a T_a + \frac{1}{32\pi G_N} \oint dy \sqrt{h}\, \bar{\varphi} \left( K - \frac{1}{\ell} \right) \\
&= -I_{\text{JT}} \ ,
\end{aligned} \tag{2.49}$$

and the BF level is identified with Newton's constant as

$$k = \frac{1}{2G_N} \ . \tag{2.50}$$

See Appendix B.1 for more details.

As before the dilaton constrains the scalar curvature to exactly $R = -\frac{2}{\ell^2}$. The simplest geometry satisfying this constraint is the hyperbolic disc, or Euclidean AdS$_2$. This is described by a metric

$$\frac{ds^2}{\ell^2} = d\rho^2 + \sinh^2\rho\, d\tau^2 \ , \tag{2.51}$$

where $\rho \in [0, \infty)$ and $\tau \sim \tau + 2\pi$. (We will discuss the cutoff disc, or "nearly AdS$_2$", as well as other topological configurations in Section 3.1 of the Discussion.) The corresponding on-shell connection describing this geometry is given by

$$A = -\sinh\rho\, d\tau\, P_0 - d\rho\, P_1 - \cosh\rho\, d\tau\, P_2 = g_\rho^{-1} g_\tau^{-1} d\,(g_\tau\, g_\rho) \ , \tag{2.52}$$

---

[7]See e.g. the heat kernel expansion of [32].

[8]Our conventions for $\mathfrak{sl}(2, \mathbb{R})$ are established in Appendix A.

where $g_\rho = e^{-\rho P_1}$ and $g_\tau = e^{-\tau P_2}$. While this expression emphasizes that $A$ is locally flat, a similar analysis to the Euclidean $dS_2$ solution reveals that it possesses a point source of flux at $\rho = 0$. That is we find, distributionally,

$$F = -\delta(\rho)d\tau \wedge d\rho\, P_2 \ , \tag{2.53}$$

which leads to an on-shell action

$$iS_{\text{BF, on-shell}} = \frac{k}{2}\varphi(0) = \frac{\varphi_0}{4G_N} \ , \tag{2.54}$$

which is the appropriate on-shell action for JT gravity on the disc (see Appendix B.1). This is depicted in Figure 3.

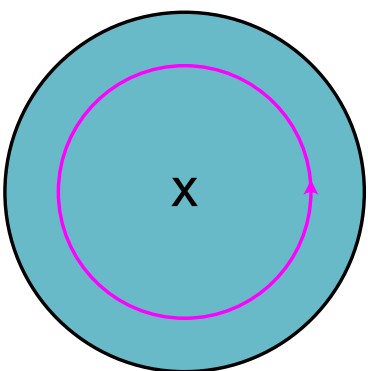

**Figure 3:** Euclidean $AdS_2$ is a hyperbolic disc. The background connection possesses a point source of flux at the origin of the disc which gives the Wilson loop in pink holonomy.

This flux also induces a holonomy of $A$ along a cycle, $\gamma$, which wraps the origin of the disc. This is easily computed from (2.52):

$$\mathcal{P}\exp\oint_\gamma A = g_\rho^{-1}\, e^{2\pi P_2}\, g_\rho \sim e^{i2\pi\mathsf{h}\, P_0} \ , \qquad \mathsf{h} = 1 \ , \tag{2.55}$$

where '$\sim$' means 'conjugable up to a periodic group element.'[9] While this conjugates to $\pm 1$ in the fundamental representation, it has a non-trivial effect on the infinite-dimensional representations describing massive matter below.

We again consider the partition function of a minimally coupled massive scalar field, $\Phi$, on this background:

$$Z_{m^2} = \det\left(\nabla_{\text{EAdS}_2}^2 - m^2\ell^2\right)^{-1/2} \ . \tag{2.57}$$

---

[9]Specifically,

$$\mathcal{P}\exp\oint A = g_\rho^{-1} e^{-i\frac{\pi}{2}P_1}\, e^{i2\pi P_0}\, e^{i\frac{\pi}{2}P_1}\, g_\rho \ . \tag{2.56}$$

Euclidean AdS$_2$ is also a maximally symmetric space admitting a set of Killing vectors, $\{\zeta_A^\mu \partial_\mu\}_{a=1,2,3}$, spanning $\mathfrak{sl}(2,\mathbb{R})$. Moreover the scalar Laplacian can be expressed as the quadratic Casimir of these Killing vectors:

$$\nabla^2_{\text{EAdS}_2} = \eta^{AB} \zeta_A \zeta_B = c_2^{\mathfrak{sl}(2,\mathbb{R})} \ . \tag{2.58}$$

Thus we conclude that $Z^2_{m^2}$ possesses poles for each weight of representation satisfying the "mass-shell" condition or **Condition 0**:

$$c_2^{\mathfrak{sl}(2,\mathbb{R})} = m^2 \ell^2 \ . \tag{2.59}$$

For highest and lowest weight representations (constructed explicitly in Appendix A), $c_2^{\mathfrak{sl}(2,\mathbb{R})}$ takes the value $j(j-1)$ and **Condition 0** implies

$$j = \frac{1}{2} \pm \nu \ , \qquad \nu = \sqrt{m^2 \ell^2 + \frac{1}{4}} \ . \tag{2.60}$$

Unlike in dS$_2$, fields on Euclidean AdS$_2$ come with Dirichlet boundary conditions at the conformal boundary $\rho \to \infty$. Only one choice of sign of $j$ will result in a normalizable fall-off consistent with this boundary condition. Without loss of generality we will suppose this occurs with the positive sign and notate $j = \frac{1}{2} + \nu$ from here onward. This is only a necessary condition: as pointed out in [42], the Dirichlet boundary condition is significantly more constraining on the normal mode spectra of Euclidean AdS in even dimensions than in odd dimensions. We will implement these constraints below, along with **Condition II**.

**Condition I** again implies that if $A$ has holonomy, $\mathsf{h}$, along a closed cycle then a state of fixed $P_0$ weight can contribute a pole when

$$\lambda \mathsf{h} \in \mathbb{Z} \ . \tag{2.61}$$

As we saw above, $A$ possesses a point source of flux at the origin of the disc which gives it holonomy for the cycle wrapping the origin, (2.55).

Lastly, unlike in the dS$_2$ context, **Condition II** is trivial for the representations considered here as every $\mathfrak{sl}(2,\mathbb{R})$ representation exponentiates to a representation of the universal cover of $SL(2,\mathbb{R})$ [43]. However as we mentioned above, normal modes consistent with the Dirichlet boundary condition are additionally constrained: the analysis of [42] shows that they fall into representations satisfying:

$$j + |n| \in -\mathbb{N}_0 \ , \qquad n \in \mathbb{Z} \ . \tag{2.62}$$

Thus $Z^2_{m^2}$ receives poles arising from weights in both highest and lowest weight representations (which, in our conventions, share the same Casimir) satisfying (2.62).

Putting this together we establish

$$Z^2_{m^2} = e^{\mathsf{P}(\nu)} \left[ \prod_{\lambda \in \mathsf{R}_j^{\text{HW}}} \prod_{n \in \mathbb{Z}} (|n| - \lambda \mathsf{h})^{-1} \right] \times \left[ \prod_{\lambda \in \mathsf{R}_j^{\text{LW}}} \prod_{n \in \mathbb{Z}} (|n| + \lambda \mathsf{h})^{-1} \right] \ . \tag{2.63}$$

From here we take the logarithm as in (2.37). We will take the extra precaution to give $\alpha$ a small imaginary part, $i\epsilon$, displacing it from its integration axis. While this is not strictly necessary for the consideration of $\text{AdS}_2$, in general, group elements of $SL(2, \mathbb{R})$ can possess imaginary holonomies and this prescription will avoid the poles associated to them. We will see this explicitly for the 'trumpet' geometry in Section 3.1. At this point we can view this as a particular regularization which results in a real answer for $\log Z_{m^2}$.

The subsequent steps follow those of the previous section. Because every weight of a lowest-weight representation is the negative of a weight of the corresponding highest-weight representation, the $\mathsf{R}^{\text{LW}}$ contribution can be mapped to $\mathsf{R}^{\text{HW}}$ with a flipped contour. Ultimately we find

$$\log Z_{m^2} = \mathbb{W}_j[A] + \mathsf{P}(\nu) \tag{2.64}$$

with

$$\mathbb{W}_j[A] = \frac{i}{2} \int_{\mathcal{C}} \frac{d\alpha}{\alpha} \frac{\cos \alpha/2}{\sin \alpha/2} \text{Tr}_{\mathsf{R}_j^{\text{HW}}} \mathcal{P} \exp \left( \frac{\alpha}{2\pi} \oint_\gamma A \right) \tag{2.65}$$

and now $\mathcal{C}$ is now the contour segments depicted in Figure 4. We can roughly regard $\mathcal{C}$ as twice $\mathcal{C}_+$, the contour segment running from $i\epsilon$ to $i\infty$, regulated by two quarter arcs around the pole at zero. We will use this regularized contour to fix $\mathsf{P}(\nu)$ below.

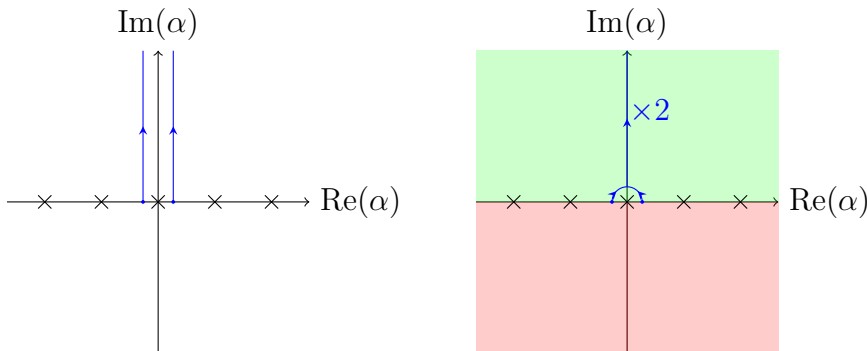

Figure 4: **Left:** The $\epsilon$-regulated contour for the $\text{AdS}_2$ spool. On shell poles are depicted as crosses. **Right:** This contour is equivalent to twice $\mathcal{C}_+$, plus two quarter-arcs avoiding the pole at zero. Damped regions of the integral are shaded in green while anti-damped regions are shaded in red.

**On-shell test**

We now take our expression to the on-shell connection describing Euclidean $\text{AdS}_2$, (2.52). This has holonomy around the origin of the disc given by (2.55). Thus

$$\text{Tr}_{\mathsf{R}_j^{\text{HW}}} \mathcal{P} \exp \left( \frac{\alpha}{2\pi} \oint_\gamma A \right) = \text{Tr}_{\mathsf{R}_j^{\text{HW}}} \left( e^{i\alpha P_0} \right) = -\frac{i}{2} \frac{e^{-i\alpha\nu}}{\sin(\alpha/2)} \ . \tag{2.66}$$

Taking $\alpha \to -i\alpha$ back to a real variable we see that $\mathbb{W}_j$ formally reproduces the one-loop determinant in its (unregulated) integral form established in [44]:

$$\mathbb{W}_j[A] = \frac{1}{2}\int_{\times}^{\infty} \frac{d\alpha}{\alpha}\frac{\cosh\alpha/2}{\sinh^2\alpha/2}e^{-\alpha\nu} = \log Z_{m^2}[\mathbb{H}_2] - \mathsf{P}(\nu) \ . \tag{2.67}$$

We assign a specific regulator to this integral through the contours depicted in Figure 4. This regulated integral is performed explicitly in Appendix C to find

$$\mathbb{W}_j = -\frac{1}{\epsilon^2} - \left(\nu^2 + \frac{1}{12}\right)\log(e^{\bar{\gamma}}\epsilon) + 2\zeta'\left(-1,\frac{1}{2}+\nu\right) + 2\nu\zeta'\left(0,\frac{1}{2}+\nu\right) \ . \tag{2.68}$$

We compare this to the minimally subtracted $\log Z_{m^2}[\mathbb{H}_2]$ obtained through heat kernel (e.g. in [42]) to fix

$$\mathsf{P}(\nu) = -\nu^2 = -\left(c_2^{\mathfrak{sl}(2,\mathbb{R})}(j) + \frac{1}{4}\right) \ . \tag{2.69}$$

We note that with this contour regularization, $\mathsf{P}$, as stated as a representation theoretic quantity, takes a unifying form in both signs of cosmological constant.

# 3   Discussion

In this paper we have investigated JT gravity in its formulation as a topological BF theory for either sign of cosmological constant. We have developed a prescription for coupling in (and integrating out) massive matter to JT gravity such that it respects the topological nature of the theory. In particular, we have shown that the one-loop partition function of a minimally coupled massive scalar field can be expressed as the integral over a Wilson loop operator of the one-form gauge field. This result is precisely the two-dimensional analogue of the Wilson spool originally developed in the context of three-dimensional gravity. We have verified that for connections describing the metrics of Euclidean dS$_2$ and Euclidean AdS$_2$, our prescription reproduces the correct one-loop determinants of a massive scalar field expressed in the 'character integral' form.

## 3.1   Quantum gravitational considerations

So far our derivation and tests of our proposal have been focussed on the on-shell geometries of Euclidean (A)dS$_2$. However, our final expression, (1.3), is expressed as a gauge-invariant and potentially off-shell functional of the connection, $A$. This allows us to insert it into the JT path-integral and investigate its expectation value in the quantum gravitational path-integral:

$$Z_{\text{JT+matter}} = \int \mathcal{D}\bar{\varphi}\mathcal{D}g_{\mu\nu}e^{-I_{\text{JT}}[g_{\mu\nu},\varphi]}Z_{\text{matter}}[g_{\mu\nu}] \equiv \int \mathcal{D}B\mathcal{D}A\, e^{iS_{\text{BF}}[A,B]}e^{\mathbb{W}_j[A]-c_2(j)-\frac{1}{4}} \ . \tag{3.1}$$

However, this theory (in either its BF or JT formulation) is pure constraint which forces the expectation value $\langle \exp \mathbb{W}_j \rangle$ to its on-shell value. This is in keeping with the statement that JT gravity with minimally coupled matter remains a solvable theory. Thus, in contrast to three-dimensional gravity, the interplay between $\mathbb{W}_j$ and quantum gravity is more indirect. Regardless, we maintain that (1.3) is, in principle, an off-shell expression and below we comment on potentially non-trivial implementations of its insertion into the JT path-integral.

**Negative curvature**

One important effect that we have neglected are boundary fluctuations. This is particularly important in the context of "nearly AdS$_2$" holography as all the dynamics arise from the Schwarzian theory at the 'wiggly' boundary determined by the dilaton boundary condition, $\bar{\varphi}(\partial M) = \bar{\varphi}_b$ [34]. While the bulk of AdS$_2$ remains fixed with $R = -\frac{2}{\ell^2}$, bulk matter back-reacts on the dilaton by modifying its equation of motion

$$\left( \nabla_\mu \nabla_\nu - \ell^{-2} g_{\mu\nu} \right) \bar{\varphi} = -8\pi G_N \, T_{\mu\nu}^{\text{matter}} \ , \tag{3.2}$$

which, in turn, modifies the location where it satisfies $\bar{\varphi}(x) = \bar{\varphi}_b$. Regardless, this effect does not change the calculation of the one-loop determinant of a free massive scalar field: this determinant is completely fixed by the bulk equation of motion and the Dirichlet boundary condition and is insensitive to how the boundary is coordinatized.[10] This is in keeping with the results of [46]. From the perspective of the Wilson spool, this insensitivity to boundary dynamics is completely natural: as long as the bulk connection remains flat, we can always deform the Wilson loop operators deep into the interior of the disc. Though we do not explore them in this paper, Wilson lines anchored to the boundary are certainly sensitive to the Schwarzian mode and lead to an exact generating functional of boundary correlation functions [9, 21].

Another, more direct, effect we can consider are contributions from new geometries to the gravitational path-integral. In Euclidean signature, the gravitational path integral admits a genus expansion that is controlled by the topological coupling $e^{\frac{\varphi_0}{4G_N}\chi}$. By topological recursion, this genus expansion can be subsequently evaluated by the 'pants construction' of a Riemann surface and integrating over all internal moduli [11]. A central ingredient in this construction is the so-called 'trumpet' geometry which is a hyperbolic two-geometry with one asymptotic boundary and one minimum length boundary with geodesic length $2\pi b$. This geometry has a metric given by

$$\frac{\mathrm{d}s^2}{\ell^2} = \mathrm{d}\rho^2 + b^2 \cosh^2 \rho \, \mathrm{d}\tau^2 \ , \qquad \rho \in (0, \infty) \ , \qquad \tau \sim \tau + 2\pi \ , \tag{3.3}$$

and is depicted in Figure 5. The corresponding flat connection is

$$A_{\text{trumpet}} = -b \cosh \rho \, \mathrm{d}\tau P_0 - \mathrm{d}\rho \, P_1 - b \sinh \rho \, \mathrm{d}\tau P_2 \ . \tag{3.4}$$

---

[10]A similar effect takes place in three-dimensional gravity: the one-loop determinant is completely fixed by Virasoro symmetry and the boundary complex structure, $\tau$ [45]. We thank Alejandra Castro for emphasizing this point.

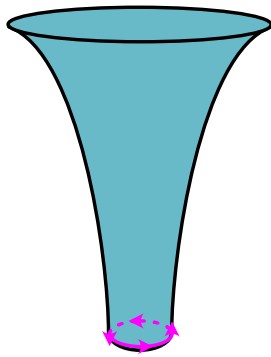

**Figure 5:** The trumpet geometry. The bottom of the trumpet is a boundary of geodesic length $2\pi b$. The Wilson loop wrapping this boundary (depicted in pink) picks up holonomy proportional to $b$.

The presence of the $\cosh^2 \rho$ in the metric (as opposed to the $\sinh^2 \rho$ for the disc) changes the character of the holonomy of the connection wrapping the compact $\tau$ cycle, $\gamma$. In particular

$$\mathcal{P} \exp \oint_{\gamma} A_{\text{trumpet}} = g_{\rho}^{-1} \, e^{2\pi b \, P_0} \, g_{\rho} \sim e^{i2\pi \mathsf{h} \, P_0} \; , \qquad \mathsf{h} = -ib \; , \tag{3.5}$$

where $g_{\rho} = e^{-\rho P_1}$ as before. As opposed to the holonomy found in Euclidean AdS$_2$, this holonomy is imaginary. Thus the expression of the Wilson spool on this geometry

$$\mathbb{W}_j[A_{\text{trumpet}}] = \frac{i}{2} \int_{\mathcal{C}} \frac{\mathrm{d}\alpha}{\alpha} \frac{\cos \alpha/2}{\sin \alpha/2} \frac{e^{-b\alpha j}}{1 - e^{-b\alpha}} \; , \tag{3.6}$$

has a significantly different pole structure than that of the disc. In particular, $\alpha$ possesses poles both along the real axis at $2\pi\mathbb{Z}$ and along the imaginary axis at $i\frac{2\pi}{b}\mathbb{Z}$.

We can imagine deforming one contour segment to wrap the (now first-order) poles along the real axis to give a spooling sum of Wilson loops wrapping the base of the trumpet arbitrarily many times

$$\mathbb{W}_j[A_{\text{trumpet}}] \supset \sum_{n=1}^{\infty} \frac{1}{n} \frac{e^{-n\pi b \nu}}{\sinh(n\pi b)} \; . \tag{3.7}$$

The matter one-loop determinant on the trumpet was also considered in [22] using the Selberg trace formula and matches the 'spooling' sum contribution of our result, (3.7). In that paper the authors also consider the matter on the 'pants' configuration and propose an all genus expansion for the JT + matter theory through duality to a random two-matrix model. We will leave adapting the Wilson spool to a full genus expansion to future work however we hope that doing so will provide new insights (e.g. into special role of the figure-eight cycle of the 'pants' geometry) and test of this duality.

Lastly, since $\mathbb{W}_j[A]$ can be interpreted as an off-shell object, we can also consider it on off-shell geometries. While off-shell geometries are typically not considered in the

gravitational path integral, particular classes have played a role in three-dimensional gravity. Two examples are locally $\text{AdS}_3$ manifolds with conical defects [47], and three-dimensional Seifert manifolds [48]. The inclusion of both examples in the path integral have been argued to solve the negative density of states in pure three-dimensional gravity. In the example of Seifert manifolds, the dimensional reduction along their $S^1$ fiber yields configurations of $\text{AdS}_2$ with defect insertions. These configurations are on-shell *almost* everywhere, however the defects induce conical singularities. The simplest example is the hyperbolic disc with a single conical deficit of $2\pi(1 - f)$ for $0 < f < 1$ (i.e. a cone) and with metric

$$\frac{\mathrm{d}s^2}{\ell^2} = \mathrm{d}\rho^2 + f^2 \sinh^2 \rho\, \mathrm{d}\tau^2 \ , \tag{3.8}$$

as depicted in Figure 6.

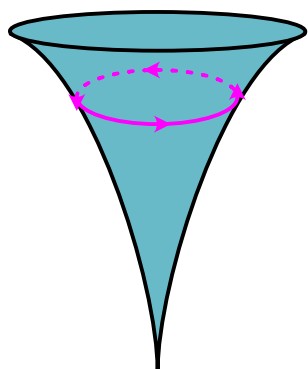

**Figure 6:** The cone geometry arising from a defect insertion in the hyperbolic disc. The conical deficit gives holonomy to the Wilson loop, depicted in pink, wrapping the tip of the cone.

From the point of view of the BF theory, this geometry is just as off-shell as the hyperbolic disc itself, which, as we saw in Section 2.2, has a unit of flux at the origin of the disc. Indeed the connection corresponding to (3.8) is a simple modification of the disc connection

$$A = -f \ \sinh \rho\, \mathrm{d}\tau\, P_0 - \mathrm{d}\rho\, P_1 - f \ \cosh \rho\, \mathrm{d}\tau\, P_2 \ , \tag{3.9}$$

which possesses holonomy

$$\mathcal{P} \exp \oint_\gamma A \sim e^{i2\pi \mathsf{h} P_0} \ , \qquad \mathsf{h} = f \ , \tag{3.10}$$

for the cycle $\gamma$ wrapping the tip of the cone (depicted in pink in Figure 6). Thus, despite being technically off-shell, the Wilson spool gives us a natural method for evaluating the scalar one-loop partition function on this geometry:

$$\log Z_{m^2}[g_{\text{cone}}] = \mathbb{W}_j[A_{\text{cone}}] = \int_\times^\infty \frac{\mathrm{d}\alpha}{\alpha} \frac{\cosh \alpha/2}{\sinh \alpha/2} \frac{e^{-f\alpha\nu}}{\sinh \frac{f\alpha}{2}} \ , \tag{3.11}$$

again up to a polynomial of the mass.

### Positive curvature

For positive curvature we have only been considering closed and compact spaces and so there are no boundary dynamics to consider. We could still consider higher-genus contributions to the path integral and the Wilson spool adapted to such backgrounds. However, it is immediately clear from the definition of the Euler characteristic

$$\chi = \frac{1}{4\pi} \int d^2x \sqrt{g} \, R = 2 - 2\mathsf{g} \,, \tag{3.12}$$

that $\mathsf{g} \geq 1$ topologies cannot be on-shell (i.e. with $R = \frac{2}{\ell^2}$). This clearly highlights the difference between the moduli space of $\mathfrak{su}(2)$ BF theory (which may host multiple flat connections on any Riemann surface) and the moduli space of JT gravity. Regardless, we may still be interested in certain off-shell solutions and the Wilson spool gives us a method for discussing matter on such solutions. As a simple example, we can consider the $S^2$ with two conical deficits, or the 'football'[11], with a metric given by

$$\frac{\mathrm{d}s^2}{\ell^2} = \mathrm{d}\rho^2 + f^2 \sin^2 \rho \, \mathrm{d}\tau^2 \,, \tag{3.13}$$

which has a conical deficits of $2\pi(1 - f)$ for $0 < f < 1$ at the North and South poles. This geometry is depicted in Figure 7. This is the positive curvature analogue of the cone discussed above, (3.8). The corresponding flat connection

$$A = i \left( f \sin \rho \, \mathrm{d}\tau \, L_1 + \mathrm{d}\rho \, L_2 + f \, \cos \rho \, \mathrm{d}\tau L_3 \right) \,, \tag{3.14}$$

possesses holonomy

$$\mathcal{P} \exp \oint_\gamma A \sim e^{i2\pi \mathsf{h} L_3} \,, \qquad \mathsf{h} = -f \,, \tag{3.15}$$

for the contour wrapping the equator (depicted in pink in Figure 7).

Thus again, despite being technically off-shell, the Wilson spool assigns an expression to the one-loop determinant of a massive scalar field on this geometry:

$$\log Z_{m^2}[g_{\text{football}}] = \mathbb{W}_j[A_{\text{football}}] = \frac{1}{2} \int_\times^\infty \frac{\mathrm{d}\alpha}{\alpha} \frac{\cosh \alpha/2}{\sinh \alpha/2} \frac{\cos(f\alpha\mu)}{\sinh \frac{f\alpha}{2}} \,, \tag{3.16}$$

up to a polynomial of the mass.

Lastly we mention that we could alternatively define the genus expansion of dS$_2$ JT gravity through its Wick rotation to Euclidean AdS$_2$ with a completely negative definite metric, as in [14, 15, 17]. In this case, topological recursion from AdS$_2$ JT gravity may yield a natural extension of the Wilson spool to this genus expansion.

---

[11]We thank Marine De Clerck for suggesting the football.

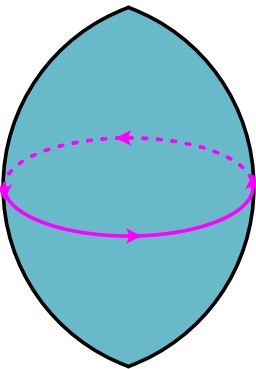

**Figure 7:** The 'football,' i.e. the sphere with conical deficits at the North and South poles. A Wilson loop around this deficit (depicted in pink) acquires holonomy $\mathsf{h} = -f$.

## 3.2 Dimensional reduction

Much of the interest and utility of JT gravity lies in its interpretation as a dimensional reduction from higher dimensional gravity. Additionally, BF theories arise naturally in the dimensional reduction from Chern-Simons theory [49], and under this reduction the three-dimensional Wilson loops have a natural intepretation of Wilson loops and defects in the two-dimensional theory [21]. In this final section, we speculate on possible relations between the construction of the Wilson spool in this paper to the three-dimensional Wilson spool constructed in [27, 28] through dimensional reduction.[12] We can sketch out a basic framework how such a dimensional reduction can work in the case of the reduction $\mathrm{dS}_3 \to \mathrm{dS}_2$, and illustrate some complications that arise. We will leave a more rigorous treatment for future work.

At the level of the action, dimensional reduction from three-dimensional Einstein Hilbert gravity to JT gravity is well understood [50]. We will denote three-dimensional objects and coordinates with a hat. For metrics static in a compact coordinate $\theta$ we can write the ansatz

$$\hat{g}_{MN}\mathrm{d}\hat{x}^M\mathrm{d}\hat{x}^N = \ell^2\varphi^2(x)\,\mathrm{d}\theta^2 + g_{\mu\nu}\mathrm{d}x^\mu\mathrm{d}x^\nu \ . \tag{3.17}$$

It then follows that, up to a total derivative,

$$\hat{I}_{\mathrm{EH}} = -\frac{1}{16\pi\hat{G}_N}\int \mathrm{d}^3\hat{x}\,\sqrt{\hat{g}}\left(\hat{R} - \frac{2}{\ell^2}\right) = -\frac{1}{16\pi G_N}\int d^2x\sqrt{g}\varphi\left(R - \frac{2}{\ell^2}\right) = I_{\mathrm{JT}} \ , \tag{3.18}$$

with

$$G_N = \frac{\hat{G}_N}{2\pi\ell} \ . \tag{3.19}$$

For our purposes it will be useful to state this as a reduction from Chern-Simons theory to BF theory. We recall that three dimensional Euclidean gravity with a positive

---

[12]We thank Watse Sybesma for suggesting and discussing this.

cosmological constant can be described by a pair of $\mathfrak{su}(2)$ Chern-Simons theories

$$\hat{I}_{\text{EH}} = -i\frac{\hat{k}}{2\pi} \int \text{Tr}\left(\hat{A}_L \mathrm{d}\hat{A}_L + \frac{2}{3}\hat{A}_L^3\right) + i\frac{\hat{k}}{2\pi} \int \text{Tr}\left(\hat{A}_R \mathrm{d}\hat{A}_R + \frac{2}{3}\hat{A}_R^3\right) , \tag{3.20}$$

where the Chern-Simons level is pure imaginary[13] and related to $\hat{G}_N$ as

$$\hat{k} = \frac{i\ell}{2\hat{G}_N} , \tag{3.21}$$

and the Chern-Simons connections are related to metric variables via

$$\hat{A}_L = i\left(\hat{\omega}^A + \hat{e}^A/\ell\right) L_A , \qquad \hat{A}_R = i\left(\hat{\omega}^A - \hat{e}^A/\ell\right) \bar{L}_A \tag{3.22}$$

where $\hat{e}^A$ and $\hat{\omega}^A$ are the three dimensional coframes and dualized spin connections, and $L_A$ and $\bar{L}_A$ generate $\mathfrak{su}(2)_{L/R}$, respectively. In order to facilitate the reduction down to a gauge theory appropriate for two-dimensions, we want to break the three-dimensional isometry algebra $\mathfrak{su}(2)_L \oplus \mathfrak{su}(2)_R$ down to an $\mathfrak{su}(2)$ subalgebra stabilizing the $S^2$.

Let $|\Sigma\rangle$ be a state that is symmetric under this $\mathfrak{su}(2)$ subalgebra, which we write as

$$\left(L_A - \Sigma_A{}^B \bar{L}_B\right) |\Sigma\rangle = 0 , \tag{3.23}$$

for some matrix $\Sigma_A{}^B$. This induces a map from $\mathfrak{su}(2)_R \to \mathfrak{su}(2)_L$ sending $\bar{L}_A \to \tilde{L}_A \equiv (\Sigma^{-1})_A{}^B L_B$. We can then design a map by hand to send the Chern-Simons theory, (3.18), to the BF theory appropriate for dS$_2$ JT gravity, (2.10) and (2.12).

For a metric of the form (3.17), without loss of generality we align $\hat{e}^3$ strictly along $\mathrm{d}\theta$ such that $\hat{e}^{1,2}$ pull back to the two-dimensional frames of $g_{\mu\nu}$ (which, to adhere to our previous conventions are $e^{0,1}$, respectively). Then $\hat{\omega}^3$ pulls back to the two-dimensional spin connection, $\omega$, and $\hat{\omega}^{1,2}$ are proportional to $\mathrm{d}\theta$. In accordance with the ansatz, (3.17), we will write $\hat{e}^3 = \ell\varphi \, \mathrm{d}\theta$ and we will additionally denote $\hat{\omega}^{1,2} = \lambda^{0,1} \, \mathrm{d}\theta$, respectively. Then under the map defined by[14]

$$\tilde{L}_{1,2} = -L_{1,2} , \qquad \tilde{L}_3 = L_3 , \tag{3.24}$$

we recognize

$$\begin{aligned}
\frac{1}{2}\left(\hat{A}_L + \hat{\tilde{A}}_R\right) &= i\left(e^0/\ell\, L_1 + e^1/\ell\, L_2 + \omega\, L_3\right) \equiv A , \\
\frac{1}{2}\left(A_L - \hat{\tilde{A}}_R\right) &= i\left(\lambda^0\, L_1 + \lambda^1\, L_2 + \varphi\, L_3\right) \mathrm{d}\theta \equiv B \, \mathrm{d}\theta
\end{aligned} \tag{3.25}$$

as the appropriate maps for a BF description of JT gravity. Furthermore it is easy to show that under this map

$$\frac{\hat{k}}{2\pi} \int \text{Tr}\left[\hat{A}_L \mathrm{d}\hat{A}_L + \frac{2}{3}\hat{A}_L^3 - \tilde{\hat{A}}_R \mathrm{d}\tilde{\hat{A}}_R - \frac{2}{3}\tilde{\hat{A}}_R^3\right] = \frac{2\hat{k}}{\pi} \int \mathrm{d}\theta \wedge \text{Tr}\left(BF\right) , \tag{3.26}$$

---

[13]Here we ignore the real part of the levels which are proportional to the gravitational Chern-Simons coupling. See [51, 52] for aspects of the dimensional reduction of this term.

[14]It is easy to verify the defines an $\mathfrak{su}(2)$ subalgebra of $\mathfrak{su}(2)_L \oplus \mathfrak{su}(2)_R$.

where we have dropped a total derivative and a $(B\mathrm{d}\theta)^3$ term which is exactly zero. Note that while its overall magnitude can be rescaled, the imaginary level of the $\mathfrak{su}(2)$ description of Euclidean dS$_2$ JT gravity is natural from the perspective of this dimensional reduction.

To establish a correspondence between the three- and two-dimensional spools, we should now investigate the behavior of three-dimensional Wilson loop operator

$$\mathrm{Tr}_{\mathsf{R}_j} \mathcal{P} \exp\left(\frac{\alpha}{2\pi} \oint \hat{A}_L\right) \mathrm{Tr}_{\mathsf{R}_j} \mathcal{P} \exp\left(-\frac{\alpha}{2\pi} \oint \hat{A}_R\right) , \tag{3.27}$$

under the restriction to the subalgebra defined by (3.23). As of yet, we do not have a full understanding of the branching rules of non-standard representations of $\mathfrak{su}(2)_L \oplus \mathfrak{su}(2)_R$ under restriction to an $\mathfrak{su}(2)$ subalgebra. In principle this could involve an integral over the highest-weight representations $\mathsf{R}_j$ weighted by an appropriate measure. If this is indeed the case, then the three-dimensional spool does not reduce to a single two-dimensional spool but an infinite family of spools!

While its interpretation is not fully clear, the origin of this complication may be related to the following. A minimally coupled scalar field in three-dimensions does not reduce to a minimally coupled field in two-dimensions but instead to a field coupled to the dilaton. For instance, the s-wave sector of a three-dimensional scalar, $\hat{\Phi}$, reduces to

$$\hat{I}_{\mathrm{scalar}} = \frac{1}{2} \int d^3\hat{x} \sqrt{\hat{g}} \left(\hat{g}^{MN} \partial_M \hat{\Phi} \partial_N \hat{\Phi} + m^2 \Phi^2\right) \rightarrow \frac{1}{2} \int d^2 x \sqrt{g}\, \varphi(x) \left(g^{\mu\nu} \partial_\mu \Phi \partial_\nu \Phi + m^2 \Phi^2\right) , \tag{3.28}$$

where $\Phi = \frac{\hat{\Phi}}{\sqrt{\pi\ell}}$. Additionally, higher Kaluza-Klein modes will result in a tower of masses which might mimic the branching of $\mathfrak{su}(2)_L \oplus \mathfrak{su}(2)_R$ representations. Better understanding the connection between Wilson spools across dimensions (in both positive and negative curvature) will be a subject of future work.

## Acknowledgements

It is a pleasure to thank Dio Anninos, Robert Bourne, Alejandra Castro, Marine De Clerck, Cynthia Keeler, Bob Knighton, Albert Law, Alan Rios-Fukelman, Watse Sybesma, and Stathis Vitouladitis for helpful conversations. Additional thanks are extended towards Alejandra Castro, Bob Knighton, Watse Sybesma, and Stathis Vitouladitis for detailed comments on a draft of this paper. This work has been partially supported by STFC consolidated grants ST/T000694/1 and ST/X000664/1, and partially by Simons Foundation Award number 620869.

# A Algebra conventions

We will take as a basis for the $\mathfrak{sl}(2,\mathbb{R})$ algebra

$$\mathfrak{sl}(2,\mathbb{R}) = \text{span}\{P_A\}_{A=0,1,2} \ , \qquad [P_A, P_B] = \varepsilon_{ABC}\,\eta^{CD}\,P_D \ , \tag{A.1}$$

with $\varepsilon_{012} = 1$ and Killing form

$$\text{Tr}(P_A P_B) = \frac{1}{2}\eta_{AB} = \frac{1}{2}\text{diag}(1,1,-1) \ . \tag{A.2}$$

It is also convenient to define $P_\pm := P_2 \pm P_1$ such that

$$[P_0, P_\pm] = \mp P_\pm \ , \qquad [P_+, P_-] = 2P_0 \ . \tag{A.3}$$

$P_\pm$ act as lowering/raising operators, respectively. The quadratic Casimir of $\mathfrak{sl}(2,\mathbb{R})$ is given by

$$\begin{aligned}
c_2^{\mathfrak{sl}(2,\mathbb{R})} = \eta^{AB}P_A P_B &= P_0^2 + P_1^2 - P_2^2 \\
&= P_0^2 - P_0 - P_- P_+ = P_0^2 + P_0 - P_+ P_- \ .
\end{aligned} \tag{A.4}$$

We construct lowest weight representation starting from a lowest weight state, $|j,0\rangle_{\text{LW}}$ satisfying

$$P_0|j,0\rangle_{\text{LW}} = j|j,0\rangle_{\text{LW}} \ , \qquad P_+|j,0\rangle_{\text{LW}} = 0 \ . \tag{A.5}$$

The representation is furnished by the action of $P_-$ which raises the weight by one. This representation is infinite dimensional with weights of the form $\lambda = j + m$, $m \in \mathbb{N}$. We can also consider highest weight states built out of $|j,0\rangle_{\text{HW}}$ satisfying

$$P_0|j,0\rangle_{\text{HW}} = -j|j,0\rangle_{\text{HW}} \ , \qquad P_-|j,0\rangle_{\text{HW}} = 0 \ , \tag{A.6}$$

and acting successively with $P_+$. Highest weight representations have weights $\lambda = -j-m$, $m \in \mathbb{N}$ and our convention has assured that both highest and lowest weight representations have the same form of Casimir:

$$c_2^{\mathfrak{sl}(2,\mathbb{R})}|j,0\rangle_{\text{LW/HW}} = j(j-1)|j,0\rangle_{\text{LW/HW}} \ . \tag{A.7}$$

The algebra $\mathfrak{su}(2)$ is given by a basis

$$\mathfrak{su}(2) = \text{span}\{L_A\}_{A=1,2,3} \ , \qquad [L_A, L_B] = i\varepsilon_{ABC}\delta^{CD}L_D \ , \tag{A.8}$$

with Killing form

$$\text{Tr}(L_A L_B) = \frac{1}{2}\delta_{AB} \ . \tag{A.9}$$

We can again define raising and lowering operators, $L_\pm := L_1 \pm iL_2$, satisfying

$$[L_3, L_\pm] = \pm L_\pm \ , \qquad [L_+, L_-] = 2L_3 \ . \tag{A.10}$$

The quadratic Casimir is given by

$$c_2^{\mathfrak{su}(2)} = \delta^{AB} L_A L_B = L_3^2 + L_3 + L_- L_+ . \tag{A.11}$$

We will also define highest weight representations starting from a state $|j, 0\rangle$ satisfying

$$L_3 |j, 0\rangle = j |j, 0\rangle , \qquad L_+ |j, 0\rangle = 0 , \tag{A.12}$$

and acting successively with $L_-$. The Casimir of the highest weight representation takes the value

$$c_2^{\mathfrak{su}(2)} |j, 0\rangle = j(j+1) |j, 0\rangle . \tag{A.13}$$

# B   JT gravity in the BF formulation

## B.1   Euclidean AdS$_2$

We begin with Euclidean JT gravity with a negative cosmological constant. The action is given by

$$I_{\mathrm{JT}} = -\frac{\varphi_0}{4G_N} \chi - \frac{1}{16\pi G_N} \int d^2x \sqrt{g}\, \bar{\varphi} \left( R + \frac{2}{\ell^2} \right) - \frac{1}{8\pi G_N} \int_\partial dy \sqrt{h}\, \bar{\varphi} \left( K - \frac{1}{\ell} \right) , \tag{B.1}$$

where

$$\chi = \frac{1}{4\pi} \int d^2x \sqrt{g}\, R + \frac{1}{2\pi} \int_\partial dy \sqrt{h}\, K = 2 - 2\mathsf{g} - \mathsf{n} , \tag{B.2}$$

is the Euler characteristic for a Euclidean manifold of genus $\mathsf{g}$ and with $\mathsf{n}$ boundary components, $h$ is the induced metric on the boundary, and $K$ is the extrinsic curvature of boundary. As is standard, we have separated off the constant part of the dilaton, $\varphi_0$, which appears now as a 'topological coupling.' The fluctuating dilaton, $\bar{\varphi}$, is non-dynamical and simply enforces

$$R = -\frac{2}{\ell^2} , \tag{B.3}$$

as a constraint. The equation of motion from varying $g_{\mu\nu}$ is

$$\left( \nabla_\mu \nabla_\nu - \ell^{-2} g_{\mu\nu} \right) \bar{\varphi} = 0 . \tag{B.4}$$

One solution and the one of primary focus in this paper is Euclidean AdS$_2$, or the hyperbolic disc:

$$\frac{\mathrm{d}s^2}{\ell^2} = \mathrm{d}\rho^2 + \sinh^2 \rho \, \mathrm{d}\tau^2 , \qquad \bar{\varphi} = \cosh \rho , \tag{B.5}$$

where $\rho \in (0, \infty)$ and $\tau \sim \tau + 2\pi$. This geometry possesses a conformal boundary at $\rho \to \infty$. The on-shell value of this solution is given by

$$\exp\left( -I_{\mathrm{JT, \, on\text{-}shell}} \right) = \exp \frac{\varphi_0}{4G_N} . \tag{B.6}$$

We now write this as a topological gauge theory. The action in question is given by

$$S_{\mathrm{BF}} = \frac{k}{2\pi} \int \mathrm{Tr}\,(BF) + S_{\mathrm{bndy}} \;, \tag{B.7}$$

where $B$ is an $\mathfrak{sl}(2,\mathbb{R})$ valued zero-form, $F \equiv \mathrm{d}A + A \wedge A$, is the field strength of an $\mathfrak{sl}(2,\mathbb{R})$ valued connection and the trace is taken in the fundamental representation.[15] $S_{\mathrm{bndy}}$ is a boundary action to make the action stationary with respect to an appropriate boundary condition. This boundary action will not play a role in our analysis, however it is standard to take

$$S_{\mathrm{bndy}} = -\frac{k}{4\pi} \int \mathrm{Tr}\,(BA) \;, \tag{B.8}$$

which is compatible with boundary condition $B - v^\mu A_\mu = 0$ (where $v^\mu$ is the unit tangent to the boundary).

To show that this is equivalent to the Euclidean JT action, (B.1), up to boundary term, we write

$$A = -(e^a/\ell)\,P_a - \omega\,P_2 \;, \qquad B = -i\,(\lambda^a\,P_a + \varphi\,P_2) \;, \qquad (a = 0, 1) \;, \tag{B.9}$$

where $\{P_A\}_{A=0,1,2}$ generate $\mathfrak{sl}(2,\mathbb{R})$ as described in Appendix A, $e^a$ are the Euclidean coframes, and $\omega = \frac{1}{2}\varepsilon_{ab}\omega^{ab} = \omega^{01} = \omega^0{}_1$ is the single independent component of the spin-connection in two dimensions. We easily compute

$$F = -T^a\,P_a - \frac{1}{2}\left(\mathrm{d}\omega + \frac{1}{2\ell^2}\varepsilon_{ab}e^a \wedge e^b\right) P_2 \;, \tag{B.10}$$

where

$$T^a = \mathrm{d}e^a + \varepsilon^a{}_b\,\omega \wedge e^b \;, \tag{B.11}$$

is the torsion two-form. Using the two-dimensional relation $\mathrm{d}\omega = \frac{1}{2}R\,\mathrm{d}\Omega$ where $\mathrm{d}\Omega = \frac{1}{2}\varepsilon_{ab}e^a \wedge e^b = d^2x\,\sqrt{g}$ is the volume form and $R$ is the Ricci scalar, we have

$$S_{\mathrm{BF}} = -i\frac{k}{8\pi} \int \mathrm{d}^2x\,\sqrt{g}\,\varphi\left(R + \frac{2}{\ell^2}\right) + i\frac{k}{4\pi} \int \delta_{ab}\lambda^a\,T^b \;. \tag{B.12}$$

We see that $\lambda^a$ act as Lagrange multipliers constraining the torsion to vanish and the resulting action is Euclidean JT action, $iS_{\mathrm{BF}} = -I_{\mathrm{JT}}$, with

$$k = \frac{1}{2G_N} \;, \tag{B.13}$$

up to boundary term. It is shown in [11, 15] that the boundary action (B.8) leads to the boundary conditions appropriate for Euclidean (near-)AdS$_2$.

---

[15]Unless otherwise stated all traces in this section are to be understood in the fundamental representation.

## B.2 Euclidean dS$_2$

We now describe JT gravity for positive cosmological constant. In order to understand it as a $\mathfrak{su}(2)$ BF theory, it will be useful to first arrive at it from Wick rotation of the Lorentzian theory. We note that this is a distinct Wick rotation from that described in [15].

The action for JT gravity for positive cosmological constant in Lorentzian signature given by

$$S_{\text{JT}} = \frac{\phi_0}{4G_N}\chi + \frac{1}{16\pi G_N}\int d^2x\,\sqrt{-g}\,\bar{\phi}\left(R - \frac{2}{\ell^2}\right)\,, \tag{B.14}$$

where

$$\chi = \frac{1}{4\pi}\int d^2x\sqrt{-g}R\,, \tag{B.15}$$

is the Lorentzian Euler character. As opposed to the previous section, we have ignored possible boundary terms for now as we will focus on spaces without boundary. Again $\phi_0$ is the constant part of the dilaton and acts as a topological coupling constant. The equation of motion for the fluctuating dilaton, $\bar{\phi}$, enforces

$$R = \frac{2}{\ell^2}\,, \tag{B.16}$$

while the equation of motion for $g_{\mu\nu}$ is

$$\left(\nabla_\mu\nabla_\nu + g_{\mu\nu}\ell^{-2}\right)\bar{\phi} = 0\,. \tag{B.17}$$

A solution of interest for this paper is the static patch given by

$$\frac{ds^2}{\ell^2} = d\rho^2 - \sin^2\rho\,dt^2\,, \qquad \bar{\phi} = \sin\rho\sinh t\,, \tag{B.18}$$

with $\rho \in (0,\pi)$ and $t \in \mathbb{R}$. This coordinate patch has two horizons at $\rho = 0,\pi$, depicted in Figure 1. We have chosen a solution for $\bar{\phi}$ that vanishes that the horizon.

We rotate to Euclidean signature by taking $t \to -i\tau$. We must correspondingly also rotate the dilaton $\phi \to -i\varphi$, which is consistent with the on-shell solution, (B.18). This leads to the $iS_{\text{JT}} = -I_{\text{JT}}$ where $I_{\text{JT}}$ is the Euclidean JT action:

$$I_{\text{JT}} = -\frac{\varphi_0}{4G_N}\chi - \frac{1}{16\pi G_N}\int d^2x\sqrt{g}\,\bar{\phi}\left(R - \frac{2}{\ell^2}\right)\,, \tag{B.19}$$

where all quantities above (the metric, Ricci scalar, and Euler character) are understood to be defined in Euclidean signature.

Under this rotation, the static patch rotates to a round two-sphere:

$$\frac{ds^2}{\ell^2} = d\rho^2 + \sin^2\rho\,d\tau^2\,, \qquad \bar{\varphi} = \sin\rho\sin\tau\,. \tag{B.20}$$

Smoothness of the horizons at $\rho = 0$ and $\rho = \pi$ require that $\tau \sim \tau + 2\pi$, commensurate with it being a coordinate on the sphere. This has an on-shell solution

$$\exp\left(-I_{\text{JT, on-shell}}\right) = \exp\frac{2\varphi_0}{4G_N} , \tag{B.21}$$

which can be viewed as the Gibbons-Hawking entropy [53] associated to the two static patch horizons and $\varphi_0$ plays the role of the horizon area.

We now write the above as a BF theory. We will start in Lorentzian signature and work with the the $\mathfrak{sl}(2,\mathbb{R})$ algebra given by (A.1). The same BF action, (B.7), also describes this Lorentzian theory with positive curvature. In this case we write

$$A = (e^0/\ell) P_0 + (e^1/\ell)P_2 + \omega P_1 , \qquad B = \lambda^0 P_0 + \lambda^1 P_2 + \phi P_1 , \tag{B.22}$$

where $e^a$ and $\omega$ are the Lorentzian coframes and spin connection. It is again easy to show that

$$F = T^0 P_0 + T^1 P_2 + \left(d\omega - \frac{1}{2\ell^2}\varepsilon_{ab}\, e^a \wedge e^b\right) P_1 . \tag{B.23}$$

which leads to

$$S_{\text{BF}} = \frac{k}{8\pi}\int d^2x\, \sqrt{-g}\,\phi\left(R - \frac{2}{\ell^2}\right) - \frac{k}{4\pi}\int \eta_{ab}\,\lambda^a\, T^b , \qquad \eta_{ab} = \text{diag}(-1,1) , \tag{B.24}$$

which is the Lorentzian JT action with $k = \frac{1}{2G_N}$.

The rotation to Euclidean signature is facilitated by $(e^0, e^1) \to (-ie_E^0, e_E^1)$ along with $\omega \to -i\omega_E$ and $(\lambda^0, \lambda^1, \phi) \to (-i\lambda_E^0, \lambda_E^1, -i\varphi)$. This sends

$$A \to (-ie_E^0/\ell) P_0 + (e_E^1\ell) P_2 - i\omega_E P_1 , \qquad B \to -i\lambda_E^0 P_0 + \lambda_E^1 P_2 - i\varphi_E P_1 . \tag{B.25}$$

(We will from here drop the subscript "$E$" with it understood that we are in Euclidean signature.) We denote the following:

$$L_1 := -P_0 , \qquad L_2 := -i P_2 , \qquad L_3 := -P_1 . \tag{B.26}$$

Then $\{L_A\}$ satisfy $\mathfrak{su}(2)$ commutation relations (A.8). We can then view $A$ and $B$ as anti-Hermitian[16] $\mathfrak{su}(2)$-valued fields:

$$A = i\left((e^0/\ell)\, L_1 + (e^1/\ell)\, L_2 + \omega\, L_3\right) , \qquad B = i\left(\lambda^0\, L_1 + \lambda^1\, L_2 + \varphi\, L_3\right) . \tag{B.28}$$

We can again map this back to gravity variables to find

$$S_{\text{BF}} \to -\frac{k}{8\pi}\int d^2x\, \sqrt{-g}\,\varphi\left(R - \frac{2}{\ell^2}\right) - \frac{k}{4\pi}\int \delta_{ab}\lambda^a\, T^b . \tag{B.29}$$

---

[16]As such their exponentiations

$$\text{e.g.} \qquad \mathcal{P}\exp\oint A \qquad \text{and} \qquad e^B , \tag{B.27}$$

are unitary group elements.

Thus in order to reproduce the Euclidean JT action, $iS_{\mathrm{BF}} \to -I_{\mathrm{JT}}$, it is necessary in this case to level to be imaginary:

$$k = \frac{i}{2G_N} \ . \tag{B.30}$$

This is reminiscent of Euclidean de Sitter gravity in three-dimensions which require imaginary levels for the Chern-Simons actions.

## C  Evaluation of on-shell integrals

Here, for completeness, we give details on evaluating the integrals appearing in the on-shell spools in Section 2.

**dS$_2$**

We will start with the integral appearing in on-shell solution for the Euclidean dS$_2$, (2.42). We will write this as

$$\mathbb{W}_j = \mathcal{W}_+ + \mathcal{W}_- \ , \qquad \mathcal{W}_\pm = \frac{1}{4} \int_\epsilon^\infty \frac{d\alpha}{\alpha} \frac{\cosh \alpha/2}{\sinh^2 \alpha/2} e^{\mp i\alpha\mu} \equiv \frac{1}{4} \int_\epsilon^\infty \frac{d\alpha}{\alpha} \, w_{\pm\mu}(\alpha) \ . \tag{C.1}$$

Note that in order for this integral converge it will be necessary to rotate $\alpha$ into the green regions appearing in Figure 2. We focus first on $\mathcal{W}_+$. The trick will be to split this integral into its UV and IR parts following [41]:

$$\mathcal{W}_+ = \mathcal{W}_+^{\mathrm{UV}} + \mathcal{W}_-^{\mathrm{IR}} \ . \tag{C.2}$$

where $\mathcal{W}_+^{\mathrm{UV}}$ contains all of the UV divergences and $\mathcal{W}_-^{\mathrm{IR}}$ converges uniformly as $\epsilon \to 0$. More specifically, $w_\mu(\alpha)$ admits a small $\alpha$ expansion as

$$w_\mu(\alpha) = \sum_{k=0}^{2} \frac{w_\mu^{(k)}}{\alpha^{2-k}} + O(\alpha) \ , \tag{C.3}$$

with

$$w_\mu^{(0)} = 4 \ , \qquad w_\mu^{(1)} = -4i\mu \ , \qquad w_\mu^{(2)} = -2\left(\mu^2 - \frac{1}{12}\right) \ . \tag{C.4}$$

We then define

$$\mathcal{W}_+^{\mathrm{UV}} := \lim_{M \to 0} \frac{1}{4} \int_\epsilon^\infty \frac{d\alpha}{\alpha} \sum_{k=0}^{2} \frac{w_\mu^{(k)}}{\alpha^{2-k}} e^{-M\alpha} \ ,$$

$$\mathcal{W}_+^{\mathrm{IR}} := \lim_{M \to 0} \frac{1}{4} \int_0^\infty \frac{d\alpha}{\alpha} \left( w_\mu(\alpha) - \sum_{k=0}^{2} \frac{w_\mu^{(k)}}{\alpha^{2-k}} \right) e^{-M\alpha} \ . \tag{C.5}$$

$M$ is an auxiliary IR regulator which will help us extract the logarithmic divergence. Ultimately we will take $M$ to zero, with all divergent in $M$ terms canceling between $\mathcal{W}_+^{\text{UV/IR}}$. The $\mathcal{W}_+^{\text{UV}}$ integral is easy to perform to find

$$\mathcal{W}_+^{\text{UV}} = \frac{1}{4}\sum_{k=0}^{1}\frac{w_\mu^{(k)}}{(2-k)}\frac{1}{\epsilon^{2-k}} - \frac{1}{4}w_\mu^{(2)}\log(e^{\bar{\gamma}}\epsilon M) , \tag{C.6}$$

where $\bar{\gamma}$ is the Euler-Mascheroni constant. To evaluate the IR contribution, we will first define a character zeta function as in [41] as

$$\zeta_\mu(z) := \frac{1}{\Gamma(z)}\int_0^\infty \frac{d\alpha}{\alpha}\,\alpha^z\,w_\mu(\alpha) , \tag{C.7}$$

for $\text{Re}(z)$ large enough to converge and then extended into the complex $z$ by analytic continuation. This admits a UV/IR split $\zeta_\mu(z) = \zeta_\mu^{\text{UV}}(z) + \zeta_\mu^{\text{IR}}(z)$ according to (C.5). $\zeta^{\text{UV}}$ is easy to evaluate as

$$\zeta_\mu^{\text{UV}}(z) = \sum_{k=0}^{2}\frac{\Gamma(z+k-2)}{\Gamma(z)}\frac{w_\mu^{(k)}}{M^{z+k-2}} \qquad \Rightarrow \qquad \frac{d}{dz}\zeta_\mu^{\text{UV}}(0) = -w_\mu^{(2)}\log M . \tag{C.8}$$

Moreover, it is easy to see that the integral in $\Gamma(z)\zeta_\mu^{\text{IR}}(z)$ is completely regular at $z=0$ and so

$$\mathcal{W}_+^{\text{IR}} = \frac{1}{4}\frac{d}{dz}\zeta_\mu^{\text{IR}}(0) = \frac{1}{4}\frac{d}{dz}\zeta_\mu(0) + \frac{w_\mu^{(2)}}{4}\log M . \tag{C.9}$$

We can massage the character zeta function as

$$\begin{aligned}
\zeta_\mu(z) &= \frac{1}{\Gamma(z)}\int_0^\infty \frac{d\alpha}{\alpha}\alpha^z\frac{\cosh\alpha/2}{\sinh^2\alpha/2}e^{-i\alpha\mu}\\
&= \frac{2}{\Gamma(z)}\int_0^\infty \frac{d\alpha}{\alpha}\alpha^z\sum_{n=0}^\infty(2n+1)e^{-(n-j)\alpha}\\
&= 2\sum_{n=0}^\infty(2n+1)(n-j)^{-z}\\
&= 4\left(j+\frac{1}{2}\right)\zeta(z,-j) + 4\zeta(z-1,-j) ,
\end{aligned} \tag{C.10}$$

where $\zeta(z,a) \equiv \sum_n (n+a)^{-z}$ is the Hurwitz zeta function and we recall $j = -\frac{1}{2} - i\mu$.

Thus adding the UV and IR contributions we find

$$\mathcal{W}_+ = \frac{1}{4}\sum_{k=0}^{1}\frac{w_\mu^{(k)}}{(2-k)}\frac{1}{\epsilon^{2-k}} - \frac{1}{4}w_\mu^{(2)}\log(e^{\bar{\gamma}}\epsilon) + \zeta'(-1,-j) + \left(j+\frac{1}{2}\right)\zeta'(0,-j) . \tag{C.11}$$

The contribution of both contours then is

$$\mathbb{W}_j = \frac{1}{\epsilon^2} + \left(\mu^2 - \frac{1}{12}\right)\log(e^{\bar{\gamma}}\epsilon) + \sum_\pm\left[\zeta'\left(-1,\frac{1}{2}\pm i\mu\right) \mp i\mu\zeta'\left(0,\frac{1}{2}\pm i\mu\right)\right] . \tag{C.12}$$

It is also instructive to evaluate this in the contour prescription given by the bottom right cartoon of Figure 2 . In this case we can write

$$\mathbb{W}_j = 2\mathcal{W}_+ + \mathcal{W}_0 + \sum_{n=1}^{\infty} \mathcal{W}_n \tag{C.13}$$

where $\mathcal{W}_0$ comes from the contribution of the half-arc wrapping the origin,

$$\mathcal{W}_0 = -\frac{i}{4} \int_{-\pi/2}^{\pi/2} d\theta \left[ \frac{\cos\alpha/2}{\sin^2\alpha/2} e^{-\alpha\mu} \right]_{\alpha=\epsilon\, e^{i\theta}} = \frac{2i\mu}{\epsilon} - i\frac{\pi}{2}\left(\mu^2 - \frac{1}{12}\right) , \tag{C.14}$$

and $\mathcal{W}_n$ is the residue around the pole at $\alpha = 2\pi n$:

$$\mathcal{W}_n = -i\frac{\pi}{2}\mathrm{Res}_{\alpha=2\pi n}\left[ \frac{\cos\alpha/2}{\alpha\sin^2\alpha/2} e^{-\alpha\mu} \right] = i\left(\frac{\mu}{n} + \frac{1}{2\pi n^2}\right)\left(-e^{-2\pi\mu}\right)^n . \tag{C.15}$$

Thus we find the alternative representation[17]

$$\mathbb{W}_j = \frac{1}{\epsilon^2} + \left(\mu^2 - \frac{1}{12}\right)\log\left(-e^{\bar{\gamma}}\epsilon\right) + 2\zeta\left(-1, \frac{1}{2} + i\mu\right) - 2i\mu\zeta'\left(0, \frac{1}{2} + i\mu\right)$$
$$+ i\mu\mathrm{Li}_1\left(-e^{-2\pi\mu}\right) + \frac{i}{2\pi}\mathrm{Li}_2\left(-e^{-2\pi\mu}\right) . \tag{C.17}$$

## AdS$_2$

The AdS$_2$ contour prescription has two contours running upwards to $i\infty$ and displaced a distance $\epsilon$ to the left and right of the imaginary $\alpha$ axis. Since there are no poles in (2.67) along the imaginary $\alpha$ axis (excepting $\alpha = 0$) we can deform this to be $2\mathcal{C}_+$ plus two quarter arcs running from $\pm\epsilon$ to $i\epsilon$, as depicted on the right side of Figure 4.

The contour segment running upwards from $i\epsilon$ to $i\infty$ is exactly the same as $2\mathcal{W}_+$ in (C.1) with $\mu \to -i\nu$ (note this is consistent with rotating it to its damped region as evidenced by the integrand (2.67)). The contribution of the two arcs are

$$-\frac{i}{4}\left(\int_0^{\pi/2} + \int_\pi^{\pi/2}\right) d\theta \left[ \frac{\cos\alpha/2}{\sin^2\alpha/2} e^{i\alpha\nu} \right]_{\alpha=\epsilon\, e^{i\theta}} = -\frac{2}{\epsilon^2} + \frac{2\nu}{\epsilon} . \tag{C.18}$$

We notice the cancellation of the linear $\nu$ term and the result is

$$\mathbb{W}_j = -\frac{1}{\epsilon^2} - \left(\nu^2 + \frac{1}{12}\right)\log(e^{\bar{\gamma}}\epsilon) + 2\zeta'\left(-1, \frac{1}{2} + \nu\right) + 2\nu\zeta'\left(0, \frac{1}{2} + \nu\right) . \tag{C.19}$$

---

[17]This is equivalent to the first representation, (C.12), via the polylogarithm identity

$$i\mu\mathrm{Li}_1\left(-e^{-2\pi\mu}\right) + \frac{i}{2\pi}\mathrm{Li}_2\left(-e^{-2\pi\mu}\right)$$
$$= -\zeta'\left(-1, \frac{1}{2} + i\mu\right) + i\mu\zeta'\left(0, \frac{1}{2} + i\mu\right) + \zeta'\left(-1, \frac{1}{2} - i\mu\right) + i\mu\zeta'\left(0, \frac{1}{2} - i\mu\right)$$
$$+ i\frac{\pi}{2}B_2\left(\frac{1}{2} + i\mu\right) + \pi\mu B_1\left(\frac{1}{2} + i\mu\right) , \tag{C.16}$$

where $B_n(x)$ are the Bernoulli polynomials.

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
