# Peer review of "Massive fields and Wilson spools in JT gravity"

_SciPost Physics_

## Round 1 · Referee Report · Anonymous (Referee 1) · 2025-6-25

Report

In this work, the author aims to rewrite the one-loop massive scalar determinant in 2D gravity as “Wilson spools”, extending their earlier work with collaborators on 3D gravity. The general programme of calculating one-loop determinants in non-trivial backgrounds is an important one. The author explains the motivations nicely and presents the calculations in a mostly clear manner. While in my opinion, this work meets the criteria for publication in the journal, I have a few concerns which should be addressed by the author:

1. The central object of this study is the holonomy along a non-contractible cycle. As in the previous works, the author attempts to justify this by noting the distributional nature of the field strength in equation (2.19). However, some of the Euclidean geometries under consideration, such as the round sphere and the hyperbolic disc, are maximally symmetric spaces and all points are isometric to one another. The singularity at the pole(s) is a coordinate artefact. Since this aspect is crucial to the whole study, it would be good if the author clarifies this point in more detail in the paper.
2. On a more technical note, it would be very helpful if the author provided a more detailed comparison with regards to the contour deformation between this work and their previous work JHEP 07 (2023) 120. The common object of study in both works is the Schwinger representation equation (2.37) of the logarithm. While in the previous work, a series of manipulations showed that the contour could be deformed into two straight lines parallel to and on two sides of the imaginary axis and a further deformation going around poles on the real axis, leading to the interpretation in terms of Wilson spools. However, in the present work, the author notes the contours are markedly different, explicitly depending on the UV regulator, and also precluding a clear interpretation in terms of Wilson spools. Despite many details on the contour, the author does not provide a physical explanation for *why* this happens, and it would be helpful to have clear statements in this regard.
3. While the rewriting of the matter determinant, while formally well-structured, looks just like a rewriting and it is unclear what new perspective it offers. In particular, it does not even look like a simplification with respect to existing works in the literature.
4. There are a few errors in numerical factors across the paper which should be corrected. For instance, there are errors in the relative factors of the boundary terms in equations (2.49) and (B.8).

As a final remark, it should be mentioned that the referencing in the paper is somewhat deficient. The author has missed some key foundational works in the literature in motivating the present work.

Recommendation

Ask for minor revision

  • validity: -
  • significance: -
  • originality: -
  • clarity: -
  • formatting: -
  • grammar: -

Author:  Jackson Fliss  on 2025-07-10  [id 5629]

(in reply to Report 1 on 2025-06-25)

I thank the referee for their careful reading and the multiple suggestions for improvement of the article. I have taken their feedback in consideration in revising the manuscript. Let me take this opportunity to address their questions:

  1. The singular points in the flux in equation (2.19) are not coordinate artefacts but instead point sources of flux that are closer to topological defects in nature. In this sense it is their existence (and not their locations in a particular coordinate patch) that are the crux of the construction. These defects are important for the consistency of the Gauss-Bonnet relation relating the integral of the Ricci scalar to the Euler characteristic (since R is the exterior derivative of the spin connection). I have added a paragraph and footnote underneath equation (2.20) emphasizing the topological nature and physical interpretation of these points of flux.
  2. The difference between the closed integration contour in d=3 and the open integration contour in d=2 for the Wilson spool is a phenomenon that is mimicked in the differences between character-integral representations of sphere partition functions between even and odd dimensions. In all cases, the open ends of the contour segments are important for reproducing logarithmic divergences that are common to even dimensional one-loop determinants. While I made statements to this effect in a previous version of my manuscript, in my revisions I have further clarified this point in the bullets starting on page 12 as well as citing the references that illustrate this contour difference more generically.
  3. For the simple examples in this paper, the rewriting is indeed not a large computational leap forward for one-loop determinants. Instead this manuscript aims to establish both a technology and a perspective on one-loop determinants that may serve useful future purposes. Firstly, realizing the effect of matter as a topological and gauge-invariant line operator may lend itself to be useful on more intricate topologies since it more directly relates to the fundamental group as well as off-shell geometries. Secondly, and more philosophically, my results situates the effective action of matter into the set of topological (and extended) operators in quantum gravity, a subject that is not fully understood but may lead to useful perspectives on gravity as an effective field theory. I have added a paragraph underneath equation (15), as well as a sentence in the discussion (underneath equation (3.7)) emphasizing the novelty and the potential utility of my result.
  4. I thank the referee for finding these; these numerical factors (as well as other minor typos) have been corrected.

“As a final remark, it should be mentioned that the referencing in the paper is somewhat deficient. The author has missed some key foundational works in the literature in motivating the present work.”

The referee is correct to point out that several key works have been excluded in the sections introducing background and motivating this work. I have substantially expanded my list of references to more comprehensively represent the important works in the field.

---

## Round 1 · Referee Report · Anonymous (Referee 2) · 2025-6-26

Strengths

1) The manuscript contains new results on minimally coupling massive matter to JT gravity with positive and negative cosmological constant by means of the 'Wilson Spool' operator

2) The treatment is rigorous and the level of technical details is appropriate

3) The paper is clear and readable

4) Concrete examples of matching with existing literature are provided, along with further directions that are open for future work

Weaknesses

No noticeable weaknesses

Report

In this paper the author develops a a method to couple massive matter to JT gravity with positive or negative cosmological constant, and compute the one-loop partition function.

By first reformulating JT gravity in terms of a BF theory, and by using the DHS formula that expresses the one loop determinant in terms of quasinormal modes, the author finds that the exact one-loop partition function can be expressed elegantly as the integral over a Wilson loop operator called Wilson spool. The latter is a collection of Wilson loops winding around closed paths of the geometry, and was introduced first in the context of 3d gravity. The author explicitly checks the new results obtained in the Wilson Spool formulation against the known partition function of a massive scalar field on Euclidean dS2 and on Euclidean AdS2, finding agreement. Several possible directions of future investigation are detailed in the conclusions, including for instance an interesting discussion on the dimensional reduction of the spool from 3d to 2d.

The paper is clear and very interesting, and considerably advances the state-of-the-art , allowing for several follow up works, since it allows the direct derivation of the exact one-loop partition function for massive fields. I recommend for publication.

I have only two questions, 1) can the author comment on whether the result be generalized to fields with different spin (here only massive scalars are treated)? 2) the author finds in formula (1.2) a polynomial term, which, as far as I understand, depends on the regularization scheme. By comparing the answer obtained from the Wilson Spool with the heat kernel one, this term seems to have a compact expression in representation theoretic terms, valid for both signs of the cosmological constant. Can the author comment on the possible physical meaning of this (i.e. was it something to be expected?)?

Requested changes

I think the paper is good for publication in the present form.

Recommendation

Publish (easily meets expectations and criteria for this Journal; among top 50%)

  • validity: high
  • significance: high
  • originality: high
  • clarity: high
  • formatting: excellent
  • grammar: excellent

Author:  Jackson Fliss  on 2025-07-10  [id 5630]

(in reply to Report 2 on 2025-06-26)

I thank the referee for their kind review and their interesting questions regarding the manuscript. Let me take the chance to address these questions here:

  1. The fields I have considered here are based on the so(1,2)=sl(2,R) representation theory of massive states. In this case, based upon the little group of a massive particle, there is no single irrep corresponding to a massive higher-spin. One can still build an action based upon a symmetric transverse traceless s-tensor, however this will break into several separate irreps of sl(2,R). In this case the application of Wilson spool should follow applying it to the analogous irreps involved. Because this point might be a distracting departure from the points of the main work I have decided not to include these comments in my revised draft.
  2. The polynomial term is not universal, but is determined both by the regularization scheme and the renormalization condition. In the spool construction these are determined by a contour choice and matching to the heat kernel, respectively. I have judiciously chosen the regularization such that P(\mu) is consistently prescribed purely by representation theory and uniformly across signs of \Lambda. The physical necessity of the inclusion of P (as opposed to in three dimensions) stems from the same necessity for contour segments as opposed to closed contours: the necessity of reproducing logarithmic divergences in even dimensions. I have expanded the discussion of this difference in even and odd dimensions in the bullet points starting on page 12.

---

## Editorial Decision

resubmitted